# Sea ice and pollution-modulated changes in Greenland ice core methanesulfonate and bromine

**O.J. Maselli[1*], N.J. Chellman[1], M. Grieman[2], L. Layman[1], J. R. McConnell[1], D. Pasteris[1], R.H. Rhodes[3], E. Saltzman[2], M. Sigl[1]**

[1] {Desert Research Institute, Department of Hydrologic Sciences, Reno, NV, USA}

[2] {University of California Irvine, Department of Earth System Science, Irvine, CA, USA}

[3] {University of Cambridge, Department of Earth Sciences, Cambridge, UK}

[*] {now at: The University of Adelaide, Australia, 5000}

*Correspondence to*:    Olivia Maselli (olivia.maselli@adelaide.edu.au)

Keywords: bromine, MSA, nitrate, sea ice, pollution, acidification, Arctic, Greenland, cryosphere

**Abstract**
Reconstruction of past changes in Arctic sea ice extent may be critical for understanding its future
evolution. Methanesulphonate (MSA) and bromine concentrations preserved in ice cores have both
been proposed as indicators of past sea ice conditions. In this study, two ice cores from central and NE
Greenland were analysed at sub-annual resolution for MSA ($CH_3SO_3H$) and bromine, covering the time
period 1750-2010. We examine correlations between ice core MSA and the HadISST1 ICE sea ice
dataset and consult back-trajectories to infer the likely source regions. A strong correlation between the
low frequency MSA and bromine records during preindustrial times indicates that both chemical species
are likely linked to processes occurring on or near sea ice in the same source regions. The positive
correlation between ice core MSA and bromine persists until the mid-20th century, when the acidity of
Greenland ice begins to increase markedly due to increased fossil fuel emissions. After that time, MSA
levels decrease as a result of declining sea ice extent but bromine levels increase. We consider several
possible explanations and ultimately suggest that increased acidity, specifically nitric acid, of snow on
sea ice stimulates the release of reactive Br from sea ice, resulting in increased transport and deposition
on the Greenland ice sheet.

## 1 Introduction

Atmospheric chemistry in the polar regions is strongly modulated by physical, chemical, and biological processes occurring in and around sea ice. These include sea salt aerosol generation, biogenic emissions of sulfur-containing gases and halogenated organics, and the photochemical/heterogeneous reactions leading to release of volatile, reactive bromine species. The resulting chemical signals influence the chemistry of the aerosol deposited on polar ice sheets. For this reason ice core measurements of sea salt ions, methanesulphonate (MSA), and bromine have been examined as potential tracers for sea ice extent (Abram et al., 2013; Spolaor et al., 2013b, 2016; Wolff et al., 2003). The interpretation of such tracers is complicated by the fact that their source functions reflect changes in highly complex systems, and signals are further modified by patterns of atmospheric transport and deposition.

MSA is produced by the atmospheric oxidation of DMS ($(CH_3)_2S$). DMS is produced throughout the world's oceans as a breakdown product of the algal metabolite DMSP, ($(CH_3)_2S^+CH_2CH_2COO^-$). DMS emissions are particularly strong in marginal sea ice zones (Sharma et al., 2012), and this source is believed to be a dominant contributor to the MSA signal in polar ice (Curran and Jones, 2000). Ice core MSA records have been used extensively in Antarctica as a proxy for local sea ice dynamics. Although the specifics of the relationship are highly site-dependent (Abram et al., 2013; Curran et al., 2003) MSA has been proven to be a reasonably good proxy for sea ice conditions (e.g., (Curran and Jones, 2000)). In the Arctic, the relationship between MSA and sea ice conditions is less straightforward due to the likelihood of multiple source regions with different sea ice conditions contributing to the ice core archived MSA (Abram et al., 2013). Until now, a significant, (r =-0.66) relationship between ice core MSA and Artic sea ice extent (specifically August in the Barents sea) has only been established for a short record from a Svalbard ice core (O'Dwyer et al., 2000). In this study we analyse the direct correlations between the MSA records from two Greenland ice core sites and the surrounding sea ice conditions in order to demonstrate the utility of MSA as a local sea ice proxy.

In this study, all dissolved or suspended bromine species are measured (including organic bromine) and shall be referred to as "bromine". The primary source of total inorganic bromine (e.g. $Br_2, Br^-, HBr$) in the marine boundary layer (MBL) is the ocean (Parrella et al., 2012; Sander et al., 2003). At concentrations of less than 0.2% that of sodium (Na), bromide ($Br^-$) makes a small contribution to ocean salinity. $Br^-$ can be concentrated in the high latitude oceans when the sea water is frozen, since the formation of the ice matrix exudes the sea-salts in the form of brine (Abbatt et al., 2012). Small, sea-salt aerosol particles blown from the surface of sea ice are typically enriched with bromine (Sander et

al., 2003) and satellite imagery has revealed that plumes of bromine (as BrO) are photo-chemically released from sea-ice zones in spring (Nghiem et al., 2012; Schönhardt et al., 2012; Wagner et al., 2001). Recently, studies have begun to link ice core records of bromide enrichment (relative to sea water $Na$ concentrations) preserved in polar ice sheets to that of local sea ice conditions (Spolaor et al., 2013a, 2013b, 2014). Spolaor and co-workers demonstrated the spring-time $Br^-/Na$ that is preserved in the ice core is a record of bromine explosion events over adjacent seasonal sea ice. A $Br^-/Na$ enrichment would therefore indicate a larger seasonal sea ice extent or conversely a shorter distance between the ice edge and the ice core site due to decreased multi-year sea ice (Spolaor et al., 2013a). However, like MSA, it is likely that the bromine – sea-ice relationship in the Arctic is complicated by the myriad of bromine source regions which influence an ice core record in addition to factors which influence the degree of enrichment of the aerosol as it travels to the ice core site. In this study we compare ice core records of bromine to those of MSA and other common MBL species in order to determine the influence of sea ice conditions and other factors on bromine concentrations.

Here we present measurements of MSA, bromine, and elemental tracers of sea salt and crustal input in two Greenland ice cores covering the time period 1750-2010 C.E.. These ice core records represent the first continuous, sub-annual resolution records of bromine in polar ice to extend beyond the satellite era. We examine the relationship between these two sea ice-modulated tracers, their relationship to independent historical estimates of sea ice distribution, and the influence of industrialization on atmospheric and ice core chemistry.

## 2   Methods

### 2.1   Ice cores

The 87 m 'Summit-2010' ice core was collected in 2010 close to Summit Station, Greenland (72°20'N 38°17'24"W, Fig. 1). The average snow accumulation at Summit, as determined from the ice core record, is ~0.22 m yr$^{-1}$ water equivalent, with few instances of melt. Due to the relatively high snow accumulation rate, seasonal analysis of the sea salt species concentrations was feasible. The 213 m Tunu core was collected in 2013 (78° 2' 5.5"N, 33° 52' 48"W, Fig. 1), approximately 3 km east of the Tunu-N automatic weather station, part of the Greenland Climate Network. The average snow accumulation at Tunu, as determined from the ice core record, is ~0.11 m yr$^{-1}$ water equivalent.

The Summit-2010 and Tunu cores were dated using volcanic horizons in sulfur (S) from well dated historic eruptions (e.g., 1815, 1835, 1846, 1854, 1873, 1883, 1912). The dating of both cores was refined by annual layer counting using seasonal cycles in Na, Ca, and the ratio of non-sea salt S/Na as described

in more detail for another Greenland ice core (NEEM-2011-S1) by Sigl et al., (2013, 2015). Annual-layer boundaries (nominal January) were defined as the minimum value in the ratio of non-sea salt S/Na following Sigl et al. (2013). The seasonal cycles in Na and Ca (from sea-salt and mineral dust emissions peaking in winter months) remain largely unaffected by rising anthropogenic emissions during the industrial period and thus can be used for annual layer counting for the entire record. The minimum in hydrogen peroxide was also used as a winter marker in the upper section of the Summit-2010 core. Timing was evaluated for consistency against other parameters including insoluble particle counts and black carbon. Monthly values were calculated assuming a constant distribution of snowfall within each year. Because of the lower accumulation rate and strong katabatic winds at the Tunu site, constraints from volcanic synchronization played a more important role in the developing the depth-age scale for the Tunu core compared with Summit-2010. First the Tunu non-sea salt S record was synchronized to the NEEM-2011-S1 volcanic record (Sigl et al., 2015) and then the required number of annual layers between volcanic horizons picked from the high-resolution chemistry.

The annual-layer dating for these ice cores resulted in a plutonium record that is consistent with other ice cores from Greenland between 1950 and 1970 and with the emission histories from nuclear weapon testing in the Northern Hemisphere (Arienzo et al., 2016). The error in the dating of the ice core records was estimated as ± 0.33 years for the Summit-2010 record and ± 1 years for the Tunu record.

## 2.2 Sampling and analysis

The ice cores were sampled from 33x33 mm cross-section sticks using a continuous melter system (McConnell et al., 2002). The silicon carbide melter plate provides three streams from concentric square regions of the ice core sample: an innermost stream (with a cross sectional area of 144 mm$^2$), an intermediate stream (340 mm$^2$) and an outer stream that was discarded along with any contaminants obtained from handling of the ice core. The innermost melt stream was directed to two inductively coupled plasma-mass spectrometers (ICP-MS, Thermo Element II high resolution with PFA-ST concentric Teflon nebulizer (ESI)) run in parallel. All calibrations and runtime standards were run on both instruments and several elements were also measured in duplicate (Na, Ce, Pb) to ensure tracking between both ICP-MS. In addition, an internal standard of yttrium flowed through the entire analytical system and was used to observe any change in system sensitivity. The instrument measuring bromine was run at medium resolution and there were no mass interferences observed at the bromine isotope mass monitored (79 amu). The sample stream was acidified to 1% $HNO_3$ to prevent loss of less soluble species, degassed just prior to analysis to minimize mixing in the sample line and sampled at a rate of 0.45ml min$^{-1}$ (McConnell et al., 2002; Sigl et al., 2013). The following elements were measured by

ICP-MS:  Br, Cl, Na, Ca, S, Ce, and Pb. Calibration of the ICP-MS was based on a series of 7 mixed standards measured at the start and end of each day for all elements except for the halides. Due to the high volatility of acid halides, a set of 4 bromine and chlorine standards were made individually in a 1% UHP $HNO_3$ matrix from fresh, non-acidified intermediate stock solution (Inorganic Ventures) every day. The intermediate melt stream was directed to a continuous flow analysis (CFA) system on which nitrate ion ($NO_3^-$) and snow acidity (sum of soluble acidic species) were measured using the technique described by Pasteris (2012) in addition to other atmospheric species of interest (Röthlisberger et al., 2000). Stable water isotopes records were also collected using the CFA system according to the method described by Maselli et al. (2013)

The analysis of MSA by batch analysis using ESI/MS/MS has been reported previously (Saltzman et al., 2006). A portion of the debubbled CFA melt stream (150 µl min[-1]) was subsampled for continuous on-line analysis of methanesulfonate by electrospray triple-quad mass spectrometer (ESI/MS/MS; Thermo-Finnigan Quantum).  This subsample was mixed with pure methanol (50 µl min[-1]) delivered using an M6 pump (syringe-free liquid handling pump, VICI).  The methanol was spiked with an internal standard of deuterated MSA ($CD_3SO_3^-$; Cambridge Isotopes) at a concentration of 52 nM. The internal isotope standard was used to correct for any changes in instrument response due to variations in water chemistry (such as acidity). The isotope standard was calibrated against non-deuterated MSA standards prepared in water from non-deuterated MSA ($CH_3SO_3^-$; Sigma Aldrich).  MSA was detected in negative ion mode using the $CH_3SO_3^-/SO_3^-$ transition (m/z 95/80) and $CD_3SO_3^-/SO_3^-$ (m/z 98/80). The concentration of MSA in the sample flow was determined from the ratio of the non-deuterated and deuterated signals after minor blank corrections. This study is the first use of the technique for ice core MSA analysis in a continuous, online mode. The uncertainty in the MSA intensity as calculated from the standard calibrations is 1%.

A second portion of the debubbled CFA melt stream was directed to an autosampler collection system to collect a discretely sampled archive of the melted ice cores. The collected samples were frozen at the end of each day and later analysed for MSA again using  ion chromatography and ESI/MS/MS.

**2.3   Calculation of anthropogenic Pb, non sea-salt S, and Br enrichment**

The Pb derived from anthropogenic sources (exPb) was calculated as the difference between total lead measure in the ice core, $[Pb]_{obs}$, and that from dust sources. The Pb from dust was calculated as a fraction of the dust proxy cerium, ($[Ce]_{obs}$).

154
$$exPb = [Pb]_{obs} - [Ce]_{obs} \times \left(\frac{[Pb]}{[Ce]}\right)_{dust}$$

153
(1)

Where the relative amount of Pb in dust, $([Pb]/[Ce])_{dust}$ , has the constant mass ratio of 0.20588

(Bowen, 1979).

Similarly the amount of non-sea salt sulfur (nssS) was calculated relative to the sea-salt sodium, ssNa:

162
$$nssS = [S]_{obs} - [ssNa] \times \left(\frac{[SO_4^{2-}]}{[Na]}\right)_{seawater}$$

(2)

Where the amount of sulfur relative to Na in sea-water, $([SO_4^{2-}]/[Na])_{seawater}$ has the constant mass

ratio of 0.252 (Millero, 1974). ssNa was calculated by comparison with calcium as both have sea salt

and dust origins (Röthlisberger et al., 2002):

$$ssNa = \frac{[Na_{obs} \times R_t - Ca_{obs}]}{[R_t - R_m]}$$

(3)

Where $R_t$ and $R_m$ are the Ca/Na mean crustal and mean marine mass ratios of 1.78 and 0.038,

respectively, (Millero, 1974).

Bromine enrichment factors relative to sea water concentrations were calculated using the following:

169
$$enrBr(Na) = \left(\frac{[Br]}{[Na]}\right)_{obs} \Big/ \left(\frac{[Br]}{[Na]}\right)_{seawater}$$

(4)

where the $([Br]/[Na])_{seawater}$ mass ratio is 0.00623 (Millero, 1974).
**2.4 Air mass back trajectories**
To identify the likely sea ice source regions of MSA and Br deposited at the ice core sites, we perform
10 day air mass back trajectories of boundary layer air masses from each ice core site using the GDAS1
archive dataset in the Hysplit4 software (Draxler and Hess, 1998). The starting height of the back
trajectories was 500 m to ensure that the monitored air masses travelled close enough to the surface at
the ice core site to potentially deposit aerosols. The vertical velocity field was taken from the
meteorological data files. Air mass back trajectories were started every 12 hours and allowed to travel
for 10 days (total number of trajectories hours = 14400 hours per month).  The number of hours that the
trajectories spent in a 2°x2° degree grid was summed over all of the trajectories for that month between
the years 2005-2013. Previous work showed that the rapid advection of MBL air was the likely source
of reactive halogens at Summit (Sjostedt et al., 2007).
**2.5   Sea Ice Correlation mapping**
In order to assess the relationships between sea ice conditions and ice core chemistry, correlation maps
were generated between annual MSA concentrations and monthly sea ice using the HadISST1 ICE
dataset at 1° latitude-longitude monthly resolution (Rayner, 2003). Pre-1979 sea ice datasets were
interpolated from sea ice extent maps compiled by Walsh (1978) which incorporate a variety of
empirical observations. The data were later bias corrected using modern satellite data (Rayner, 2003).
Correlations were performed separately for the satellite period (1979-2012) and for the extended record
(1900-2012), excluding the period 1940-1952 when the record has no variability due to scarcity of data
(Rayner, 2003). Because strong DMS emissions occur in marginal sea ice zones (Sharma et al., 2012),
we considered both sea ice concentration (SIC) and the area of open water in the sea ice pack (OWIP)
which represents the size of the marginal sea ice zone. OWIP is defined as the difference between sea
ice area (calculated from sea ice concentration over the area of the grid cell) and sea ice extent (NSIDC).
A SIC of 15% was used as the threshold for a grid cell to contribute to sea ice extent. The area of OWIP
was calculated within the coastal areas as defined by the results of the air mass back trajectories (Sect.

3.4).

Outliers were removed from the MSA time series (see Fig. 2) before the correlations were performed.
The outliers were removed using the technique described by Sigl (2013) for identifying volcanic signals
using a 25 year running average filter. Correlations were performed on an annual rather than seasonal
basis because the seasonality of ice core MSA is distorted due to post-depositional migration of MSA
signal at depth in the snow pack (Mulvaney et al., 1992) (Fig. 3, S1).

**3   Results**
**3.1   Bromine**
Ice core measurements of bromine at Summit and Tunu covering the period 1750-2010 are shown in
Fig. 2.  Ice core Br levels at each site were stable until ~1820 at Summit and ~1840 at Tunu when they

both decreased by ~1 nM, establishing a new baseline that was stable until the mid 1900s. Both ice cores also show a Br peak in the late 20[th] century. The concentration values and the timing of inflections in concentrations were determined by a 3 step linear regression of the data set. The analysis was performed by simultaneous linear least squares fitting of 3 straight lines joined by 'inflection points' to the data set. The variables of the fitting procedure were the slopes and intercepts of each line as well as the x-axis locations at which the total function switched from one linear section to the next (the inflection points). Initial guess values were supplied for each variable to help the fitting procedure reach reasonable values. A summary of the regression results can be found in Table S1.

Sea-salt transport onto the Greenland ice sheet occurs predominantly during winter. Historically the winter-time sea-salt maximum was believed to be due to increased cyclonic activity over the open oceans (Fischer and Wagenbach, 1996) though more contemporary studies show that blowing snow from the surface of sea-ice may be a significant source (Rankin et al., 2002; Xu et al., 2013; Yang et al., 2008, 2010). At Summit, a winter-time maximum is observed in the most abundant sea salts, Na and Cl (Fig. 3). Bromine also shows a significant winter-time signal, however the annual maximum appears in mid-summer - at concentrations ~70% above winter levels (Fig. 3a). Comparison with Br measured in weekly surface snow samples collected from Summit (from 2007-2013; GEOSummit project) confirms that this summer signal is real and not a result of post-depositional modification of seasonality of the bromine signal (Fig. S2). The results from that study confirm that total Br concentrations peak in summer on the ice sheet closely following the Br cycle observed in the Summit-2010 ice core. In addition to the comparison with the Geosummmit data, in the ice cores studied here there are routinely more than 10 measurements made within a yearly layer of snow giving confidence to the allocation of a summer maximum in bromine at Summit. Analysis of the annual cycle of bromine in the Tunu ice core also shows a summer maximum when averaged over the entire ice core time series but with significantly larger error than observed at Summit. The timing of this peak suggests a predominant summer-time deposition of bromine that dwarfs that from winter sea salt sources.

The shape of the annual bromine cycle does change slightly over the course of the Summit record (see Fig. 3). Starting in the early 1900s the annual bromine cycle slowly becomes broader. A slight shift in the maximum from a solely summer peak in the preindustrial era towards a broad summer-spring peak by 1970 is observed (Fig. 3 lower plot). Comparison with the sea salt tracer, sodium, which does not undergo the large temporal shift and broadening of its seasonal cycle shows that this change in bromine seasonality is not linked to changes in production or transport of sea-salt aerosols or even dating uncertainties in the ice core but perhaps the introduction of an additional, smaller bromine source in the spring-time during the industrial era.

Both ice cores show a predominantly positive Br enrichment throughout the year (Fig. S3, S4) relative to both sea salt elements chlorine and sodium. This enrichment reaches a maximum in mid to late summer at Summit (Fig. 3). We assume that this enrichment reflects Br enrichment in the aerosol transporting Br to the ice sheet. In a comprehensive review of global aerosol Br measurements, Sander et al. (2003) concluded that in general, aerosols which showed positive Br enrichment factors were of sub-micrometer size. These small aerosols can travel further (lifetimes of around 5-10 days) and due to their larger surface/volume ratio may experience more atmospheric processing than larger aerosols, resulting in the positive enrichment. However, post-depositional reduction of the bromine concentration is a possibility during the summer months due to photolytic processes at the snow surface. This may be the cause of the noisiness of the bromine signal within the lower accumulation, Tunu core. However, the increased snow accumulation that occurs during the summer months in both central and northern Greenland (Chen et al., 1997) should act minimise these bromine depleting effects driven by increased insolation in summer and indeed Weller (2004) has shown that accumulation rates of this size are large enough to prevent the post-deposition loss of other species such as nitrate and MSA.

Both sites also show a (small) positive enrichment of chlorine relative to sodium, which is amplified at small sodium concentrations. Chlorine containing aerosols are expected to undergo similar chemical processing to bromine containing aerosols but the enrichment factors of bromine (relative to sodium) are much larger which is likely due to the high solubility of bromine species such as HBr (Sander et al., 2003) . Alternatively, the chlorine enrichment could be interpreted as a sodium depletion of the aerosols particularly in those of small diameter where both concentrations are low; this would amplify the bromine enrichment (relative to sodium) but would not explain the bromine enrichment relative to chlorine. It is likely that both halogens undergo some degree of enrichment and the sodium undergoes some depletion in the aerosols though it is difficult to determine this from the data.

A summer-time maximum in Br enrichment was also observed by Spolaor (2014) in a short segments of Antarctic Law Dome ice core as well as two Arctic ice cores. Spolaor et al. believe that the main source of the inorganic bromine originated from spring-time bromine explosion events above sea ice and the summer-time maximum could possibly be an indication of lag-time between bromine containing particles becoming airborne and their deposition. Further investigation is needed to definitively establish the seasonality of bromine deposition at the poles. However the results of the Arctic ice cores studied here suggest that the summer maximum in bromine deposition is indeed real.

In the Tunu ice core, 11% of the monthly bromine enrichment measurements relative to Na were negative (less than the Br/Na seawater ratio, Fig. S3) and 12% were negative relative to Cl. It is possible

that the negative enrichment values observed in the Tunu ice core are therefore a result of larger aerosols (> micrometer) reaching the site due to its proximity to the coast (and thus the likely sea ice aerosol source region) in comparison to Summit.

## 3.2  MSA

The Summit-2010 MSA record (Fig. 2) replicates that measured by Legrand in 1993 (Legrand et al., 1997) and extends it an additional 17 years (see Fig. S5). The mean Summit-2010 MSA measurements over the period 1984-1992 (2.0±0.7 (1$\sigma$) ppb) also compare well with the results of the sub-annually sampled Summit snow pit study performed by Jaffrezo et al., (1994); 2.1±1.8(1$\sigma$) ppb. Both the Legrand and Jaffrezo studies measured MSA using ion chromatography of discretely sampled snow and ice. The similarity between the Summit-2010 measurements and the results of these studies demonstrates that the new, continuous technique is able to achieve a comparable accuracy in MSA measured concentrations to the traditional, discrete technique. It also demonstrates that negligible amounts of MSA are being lost by using the continuous melt method.

The Tunu measurements represent the first MSA profile at this location. Replicate measurements of the entire Tunu ice core were performed with the on-line, continuous technique by melting a secondary stick of ice cut from the original Tunu ice core. The replicate measurements closely followed the original MSA measurements demonstrating the reproducibility, stability and high precision of the continuous MSA technique (Fig. S6). The Tunu MSA record was also reproduced using discrete samples collected from the CFA system (Fig. S7).

At Summit, MSA concentrations averaged 48 nM in the late 18$^{th}$ century, compared with just 27 nM at Tunu. From 1878-1930 MSA concentrations at Summit plateaued at 36 nM after which they began to drop rapidly, at a rate of 0.27 nM/year, reaching 18 nM by 2000 C.E.. Large fluctuations in the MSA record after this time make it difficult to assess the most recent trend in Summit MSA concentrations. MSA concentrations in the Tunu core showed a similar temporal variability to those in the Summit record, and until the mid-20$^{th}$ century, were consistently lower in magnitude. MSA concentrations only began to decline consistently at Tunu after 1984, almost 50 years after the rapid decline observed in the Summit record. After 2000 C.E., large fluctuations in concentration were again observed making the modern-day trend in MSA concentration at Tunu difficult to establish.

Comparison with the total sulfur record (Fig. 4) reveals that during the preindustrial period, MSA contributes to ~12% and ~ 7% of the total sulfur signal at Summit and Tunu, respectively, compared with < 2% at the height of industrial period (1970 C.E.) at both sites.

The low frequency, preindustrial trend in MSA concentrations seen in these ice core records closely follows that of bromine; particularly distinct is the decrease in both MSA and bromine at both sites in the early to mid 1800s (Tables S1 and S2). In the 1900s, however, both sites show a divergence between the MSA and Br records—as MSA begins to decline, Br concentrations increase.

A dramatic shift in the 'timing' of the annual MSA maximum in Summit-2010 ice core is illustrated in Figs. 3c and S1. The signal shifts gradually and continuously along the length of the the entire Summit-2010 record from a spring to winter maximum (Fig. S1). This phenomenon has previously been observed in several Antarctic ice cores and has been attributed to post-depositional migration within the ice due to salt gradients (Mulvaney et al., 1992; Weller, 2004). At very low accumulation ice core sites post-depositional loss of MSA (and nitrate) must also be considered. Extrapolation of data collected by Weller (2004) from a series of East Antarctic ice cores predicts that sites with annual average accumulations of greater than 105 kg m$^{-1}$ yr$^{-1}$ (0.105 m yr$^{-1}$) will not show post-depositional loss of MSA (or nitrate). Both ice cores in this study have sufficient average annual accumulation that post-depositional loss of MSA (and nitrate) is predicted to be negligible and so is not discussed further.

## 3.3 Acidic Species

In winter, with the collapse of the polar vortex, polluted air masses enter the Arctic region as the phenomenon known as the Arctic haze (Barrie et al., 1981; Li and Barrie, 1993). $SO_2$ and $NO_x$ from the haze are adsorbed onto aerosols or deposited directly on the ice/snow and oxidised to sulfuric ($H_2SO_4$) and nitric acid ($HNO_3$). There are also natural sources of $SO_2$ (biomass burning, volcanic eruptions, oceans (Li and Barrie, 1993; McConnell et al., 2007; Sigl et al., 2013) and $NO_x$ (microbial activity in soils, biomass burning, lightning discharges (Vestreng et al., 2009) as well as other snow/ice acidifiers including MSA, hydrogen chloride and organic acids released from biogenic or biomass burning sources (Pasteris et al., 2012).

The annual cycle for nitrate ($NO_3^-$) is shown in Fig. 3d. Before 1900 C.E. the nitrate shows a seasonal maximum in late summer/early fall after which the maximum shifts to late spring/early summer. Although there are biological sources of nitrate in the ice core aerosol source regions, in a recent study focused on the $NO_3^-$ and $\delta^{15}N - NO_3^-$ record in the Summit-2010 ice core, Chellman et al. (2016) concluded that the preindustrial (1790-1812 C.E.) $NO_3^-$ seasonal cycle was driven by biomass burning emissions. However, in the modern era (1930-2002 C.E.) oil-burning emissions became the dominant source of $NO_3^-$ in the snow-pack. The change in the dominant $NO_3^-$ source due to industrialisation is the cause of the shift in timing of the seasonal cycle.

Total snow acidity was stable at both sites from 1750 through to ~1900 C.E. except for sporadic, short-lived spikes due to volcanic eruptions. The average preindustrial acidity was the same at both sites (~1.8 μM). Both records also show two distinct maxima in acidity centred on 1920 and 1970 C.E. (Fig. 4) with Tunu displaying higher acidity than Summit over the entire industrial period. Overlaid with the acidity is the total sulfur (S) record for both ice cores. The high correlation between the acidity and S records illustrates that the sulfur species are the dominant natural and anthropogenic acidic species in the ice cores. The trend in acidity closely follows the global $SO_2$ emissions with maxima from coal (~1920 C.E.) and coal plus petroleum combustion (~1970 C.E.), respectively (Smith et al., 2011). After 1970 the records of acidity and S deviate. This deviation can be attributed to the presence of nitric acid that remains at a relatively high concentration in the late 20th century whilst sulfur species reduce in concentration (Fig. 4).

$NO_3^-$ concentrations show no trend during the preindustrial era in either ice core records, averaging 1.1(±0.02) μM and 1.3(±0.03) μM for Summit and Tunu, respectively. The higher signal-to-noise ratio in the Summit-2010 record reveals a small peak in $NO_3^-$ concentrations centred on ~1910. The Tunu record also shows elevated $NO_3^-$ concentrations over this period. However the large variability in the signal makes it difficult to establish a higher resolution temporal trend. Both records clearly show a large increase in $NO_3^-$ after 1950, peaking in ~1990 and followed by a general decreasing trend with the average $NO_3^-$ levels still double that of preindustrial concentrations: 2.1μM and 2.3 μM at Summit and Tunu, respectively.

The nitrate records from both sites follow the trend in northern hemisphere $NO_x$ emissions with a peak in ~1910 and 1990 C.E.– a result of emissions from increases in both Northern Hemisphere fertilizer usage and biomass and fossil fuel combustion (Felix and Elliott, 2013).

## 3.4 Air mass back trajectories

Air mass back trajectory results demonstrate that air masses reaching the Summit-2010 site between March and July originate primarily from the South/South-East of the ice core site (Fig. 5a). Previous back trajectory analyses by Kahl *et al*. (1997) also linked individual spikes in their Summit MSA record to air masses that had passed over this same region of coast (SE Greenland) within the previous 1-3 days. Similar back trajectories were calculated for Summit-2010 up to heights of 500 and 10,000m (total column trajectory, Fig. 5a, S8a) illustrating that air masses that travel in the free troposphere and lower troposphere follow similar back trajectories and likely share the same source regions.

The results for Tunu indicate that air masses arrive primarily from the west coast of Greenland, passing
over the Baffin Bay area, but there is also significant contribution from both the SE and NE (in May)
coastal areas (Fig. 5b, S8b). Of these two secondary areas it is likely that aerosols transported from the
NE would have a greater influence on the ice core concentrations due to proximity to the ice core site.
Aerosol deposited at Tunu therefore represents a mixture of source regions, but are likely dominated by
the NW Greenland, Baffin Bay coastal region.

## 3.5   MSA - Sea Ice correlations

Locations which showed a sea ice concentration (SIC) variability greater than 10% (the average
estimated range of uncertainty in the satellite measurements) and have a significant correlation to MSA
(t-test, $p<0.05$) are displayed in Figs S9 and S10 for the months of March-July. A greater weight must
be placed on the post-1979 sea ice concentration maps as these were derived from passive microwave
satellite data and, where available, operational ice chart data.  The likely air mass source regions, as
defined by the results of the air mass back trajectories, are indicated by the black bordered regions.
Within these areas there is generally a negative correlation between SIC and MSA, particularly in the
spring months and only small patches that show large correlation ( $>0.4$). The large areas of positive
correlation along the east coast and in the western Barents Sea are striking for the Summit-2010 record,
however, these areas are outside of the defined air mass source region and thus are unlikely to be
contributing to the ice core aerosol records. The positive correlation is likely an artefact of the negative
autocorrelation between sea ice conditions in this region and the SE coast source region (Fig. S11).
The effect of the estimated error in dating of the MSA records on the SIC correlation maps is explored
in Fig. S12. By shifting the dating of the MSA records to either extreme of the dating error estimate and
replotting the SIC correlation plots it is clear the error in the dating of the MSA records does not affect
the sign of the correlations displayed on the maps but can have an affect on the magnitude of the
correlation found in different locations. This is likely a result of the peaks in the MSA record being
shifted in or out of temporal coherence with peaks in SIC at the different locations.
Over the period 1900-2010 C.E. highly significant correlation (t-test, $p<0.001$) is found between the
annual ice core MSA and the amount of open water in the ice pack (OWIP, representing the area of the
marginal sea ice zone, Figs. 6a and 7a; lower plots) in these aerosol source areas. For both ice cores the
source region OWIP trend is followed by the MSA. In the Summit-2010 ice core the highest correlation
between annual MSA and monthly OWIP occurs in May ($r=0.58$, $p<0.001$) though the following months
through to July all show highly significant correlations (July $r=0.53$, $p<0.001$). For comparison, the May

SIC correlation map is also shown as the upper plots in Figs. 6a. Figs. 3f and S13 demonstrate that this time period (May-July) corresponds to the peak and then rapid decline in the amount of annual OWIP within the Summit-2010 aerosol source area because of the decreasing extent of sea ice. Rapid loss of sea ice reveals areas of biological activity previously capped by the ice allowing surface-atmosphere exchange of DMS, resulting in the seasonal peak in atmospheric MSA correlation with the peak in the area of OWIP.

At Tunu the highest correlation over the 1900-2012 C.E. period is found between annual MSA and annual OWIP (r=0.59, p<0.001), though the July OWIP shows the highest monthly correlation and is also highly significant (r=0.41, P<0.002). For comparison, the July SIC correlation map is also shown as the upper plots in Figs. 7a. Due to the more northerly location of the Tunu aerosol source region, the sea ice pack in this region is generally less fractured and break-up occurs later in the year, with a sharp peak in OWIP occurring in July (Fig. S13). The higher stability of the ice pack throughout the year compared to that in the Summit-2010 source region is the likely reason the Tunu MSA shows highest correlation with the annual average of the OWIP. However, like Summit-2010 the highest monthly OWIP correlation occurs between the annual MSA and the timing of the maximum in annual OWIP (July).

Over the shorter, satellite era (1979–2012 C.E.) again Tunu shows strongest correlation between annual MSA and annual OWIP though at a much lower significance (r=0.32, p<0.05), and the highest monthly correlation occurs in March (r =0.2, p<0.1) albeit with low significance. The significance of the Tunu correlation over this period can be dramatically increased (annual OWIP $r$ =0.54; $p$<0.001, March OWIP r=0.63, p<0.001) if the closer, secondary aerosol source region (NE Greenland, 80°−73°N, 20°−0°W) is assumed to also influence the site in equal proportion. March corresponds to the timing of increased insolation and thus the rapid increase in ice algal production (Leu et al., 2015). The shift from a July to March peak in the correlation of OWIP with annual Tunu MSA may be a result of the reduced overall SIE (and thus OWIP) influencing the timing of MSA production. Unfortunately, the post-depositional migration of the MSA signal within the ice cores masks any evidence of true seasonal MSA shifts. Summit-2010 also shows a much less significant monthly OWIP correlation with the annual MSA signal over this time period, with the most significant correlation again occurring in March (r =0.4, p<0.02). The greater significance of both the SIC-MSA and OWIP-MSA correlations at both sites over the longer time period is likely a result of the averaging of any MSA production or transport variability as well as the dominance of the low frequency variability of both time series on the overall correlation.

## 3.6 MSA and bromine relationship

In an era where climate is driven by only natural forcings, chemical species that share a common source should show broadly consistent variability. This is evident in the preindustrial section of both ice core records where the relationship between MSA and Br (monitored as Br/MSA) remains constant over the entire period (Fig. 4) despite individual records going through step function changes. Using a 25 year running average on all records, the correlation between MSA and Br over the preindustrial period was calculated as: Summit-2010: r=0.282 (p=0.0008); Tunu: r= 0.298 (p = 0.0004), n= 138. After ~1930 C.E., relative increases in Br concentrations cause the Br/MSA ratio to increase above the stable preindustrial levels by more than 160%, reaching a peak in ~2000 C.E. at both sites.

Bromine in excess of what is expected from a purely sea ice source (non sea ice bromine, nsiBr) was calculated by comparison to the other sea ice proxy, MSA. A linear regression of MSA versus Br was performed with the preindustrial data (1750-1880 C.E.) to establish the relationship between the two proxies during an era free of anthropogenic forcing (Figure S14a,b). This relationship was then extrapolated into the period after 1880 C.E. in order to estimate the amount of bromine sourced only from sea ice sources during the industrial era. The MSA record was smoothed with a $9^{th}$ order polynomial function before being used in the extrapolation to reduce the noise in the resultant record whilst maintaining the low frequency trends (Figure S14c,d). nsiBr is thus the difference between the total bromine measured and the calculated, natural sea ice bromine (Figs. 8 and S14e,f); in contrast to $Br_{exc}$ defined by Spolaor (2016) as the amount of bromine in excess of the Br/Na seawater ratio.

An estimate of the nsiBr is shown in Figs. 6,7 and 8. By definition, nsiBr is essentially constant during the preindustrial period, but during the industrial period nsiBr peaks, reaching a broad maximum between 1980-2000 C.E. of ~3.4nM and 1.9nM at Summit and Tunu, respectively.

## 4 Discussion

The significant correlation between variability of marginal sea ice zone (OWIP) area within the identified source regions and the MSA records suggests that MSA records can be used as a proxy for modern sea ice conditions in these areas. North Atlantic Oscillation (NAO) proxy records developed in Greenland ice core records (Appenzeller et al., 1998) suggest that although the northern hemisphere climate phenomenon has shown variability over the past 200 years, its effect is damped in Northern Greenland (Appenzeller et al., 1998; Weißbach et al., 2015) so we can assume that no major changes in atmospheric circulation patterns have occurred to change the source regions for the marine aerosols between the preindustrial and industrial periods. If this assumption is true, our identification of MSA as a sea ice proxy (specifically a marginal sea ice zone proxy) may be valid for time periods both before

and after 1850 at each ice core site.
The MSA records reveal that after 1820 C.E. a gradual decline in sea ice occurred along the southern
Greenland coast (reflected in the Summit-2010 core) and that this decline in sea ice did not extend
significantly to the most northern Greenland coastline (reflected in the minimal change in Tunu MSA
during this period). It is not unexpected that the Summit-2010 record would show the most dramatic
changes in sea ice since we have demonstrated that the Summit sea ice proxy (MSA) is sourced from
the south-east Greenland coast – an area sensitive to climate changes as it is primarily covered by young,
fragile sea ice. The timing of the sea ice decline is coincident with the end of the Little Ice Age, identified
from $\delta^{18}O$ ice core records as spanning the period 1420-1850 C.E. in Greenland (Weißbach et al., 2015).
The dramatic dip in sea ice reflected in both the Tunu MSA and Br records at 1830 C.E. (and also seen
less dramatically in Summit) also appears in the multi-proxy reconstruction of sea ice extent in the
Western Nordic Seas performed by Macias Fauria et. al. (2010).  This may be evidence of a 1830 C.E.
sea ice decline event isolated to the east Greenland coast as the ice core records do not replicate the
other dramatic, early 20[th] century fluctuations observed in the latter part of the Western Nordic Seas
reconstruction.
From the ice core records it appears that the greatest decline in Greenland sea ice began in the mid 20[th]
century, dropping to levels that are unprecedented in the last 200 years.  This decline is observed along
the entirety of the Greenland coast. Sea ice declined first around the southern coast (from 1930 C.E.,
reflected in Summit-2010) followed 54 years later by the more northern coastline (reflected in the Tunu
record, see infection timings in Table S1). This sea ice decline is coincident with the sustained increase
in greenhouse gases which has been identified as the major climate forcing and driver of increased
global temperatures during the 20[th] century (Mann et al., 1998) and follows the same general trend in
Arctic wide sea ice extent observed by Kinnard (2008).
Bromine (more specifically bromine enrichment (Spolaor et al., 2014) and bromine excess (Spolaor et
al., 2016)) has also been suggested as a possible proxy for sea ice conditions, however the timing of the
largest bromine aerosol deposition, in summer, does not coincide with the largest growth or extent of
new sea ice. Sea ice begins to increase only at the end of summer as the fractures in the ice cover are
re-laminated and the ice edge begins to advance southward (see Fig. 3f). Fig. S4 compares the record
of total bromine and bromine enrichment (calculated relative to sodium, enrBr(Na)) from the Summit-
2010 ice core. The major discrepancies between the two records occur when the total sodium signal has
sharp maxima causing dips in the enrBr(Na) record in ~1954 and 1990 C.E. and the magnitude of the
low frequency variability in enrBr(Na) is not as great as in the total bromine record. This is also

demonstrated in figs. 6 and 7 where the enrBr(Na) records are compared with the OWIP records. Whilst both series share high frequency temporal features, over the longer term (1900-2010) the low frequency trend is dramatically different. We are not discounting enrBr(Na) as a viable proxy for sea ice conditions, however the use of Na to try and extract the pure sea water component of the Br is complicated by the fact that a lot of Na comes from the sea ice surface as well as from the open ocean. Na itself has been used as a sea ice proxy in several prominent studies (Wais Divide Project Memebers, 2013; Wolff et al., 2003) because, like Br, Na is incorporated into the snow on the surface of the sea ice and can be subsequently blown aloft to produce the atmospheric Na signal seen in the ice core. In addition, the Na concentration is fractioned upon the formation of the ice when mirabilite ($Na_2SO_4$) is precipitated out of the brine solution at -8°C (Abbatt et al., 2012).

The calculated, non-sea ice bromine records (nsiBr) for both ice cores are shown in figs. 6 and 7. Like the enrBr(Na) records, the nsiBr records share some of the high frequency features of the OWIP records, however there is no significant correlation between nsiBr and the selected OWIP records over the short time period. This supports the supposition that the nsiBr record is indeed an extraction of the non-sea ice component of bromine from the total bromine record. Over the longer time period there is a significant negative correlation between OWIP and nsiBr at both sites (Summit-2010: $r=-0.7$, $p<0.001$, Tunu: $r=-0.22$, $p<0.02$). This result is likely an artifact of the positive correlation from the MSA records used to generate the nsiBr records.

So what is the summer-time source of bromine? What is the cause of the increase in spring-time bromine explosion events in the industrial era? (see Fig. 3, lower panel) and why does the bromine record deviate from the sea ice proxy record (MSA) around the same time? Possible sources of bromine and the factors which may effect the resultant bromine deposition flux  are discussed below.

## 4.1    Alternate sources of bromine

## 4.1.1  Combustion of coal

Bromine is present in coal (Bowen, 1979; Sturges and Harrison, 1986) and coal burning is therefore a potential source of increased bromine deposition on the Greenland ice sheet over the period 1860-1940 (McConnell and Edwards, 2008).  McConnell et al. (2007) demonstrated that pollution from the Northern American coal burning era was deposited all over Greenland leaving as its fingerprint large amounts of black carbon and toxic heavy metals. Sturges (1986) measured the relative concentrations

of Br and Pb in particulates emitted from the stacks of coal fired power stations and found a molar ratio (Br:Pb) ranging between 0.36-0.67:1. Figure 8 illustrates that at both Summit and Tunu the exPb (lead not from dust sources) preserved in the ice cores over the coal burning era (~1920) was less than 1nM. This concentration implies that the upper limit to the amount of bromine deposited from coal combustion would be 0.67nM (assuming no loss of bromine from the particulates during transportation). This is an insignificant amount compared to the total Br signal preserved in the ice at this time. Coal combustion is not the major cause of the elevated industrial Br concentration.

### 4.1.2 Leaded Gasoline

The largest global, historical, anthropogenic source of bromine is thought to be the combustion of leaded gasoline. Large quantities of 1,2-dibromoethane (DBE) were added to leaded fuel as a scavenger for Pb preventing lead oxide deposition by converting it to volatile lead bromide salts as well as $CH_3Br$ (Berg et al., 1983; Nriagu, 1990; Oudijk, 2010). In 1925 C.E. gasoline had a Br:Pb molar ratio of 2:1 in a formulation which is now called "aviation fluid'. The Br:Pb molar ratio was reduced to 1:1 in the 1940s except in places such as the Soviet Union which continued to use "aviation fluid" for motor gasoline (Thomas et al., 1997). Although the consumption of leaded gasoline has been well documented, particularly in North America, the estimates of the emissions of bromine compounds from the combustion process are still unclear. Estimates of the amount of DBE that is converted into gaseous $CH_3Br$ range from 0.1% to 25% (Bertram and Kolowich, 2000) and direct measurements of exhaust fumes across NW England found a Br:Pb ratio of between (0.65-0.8):1 in the airborne particulates (Sturges and Harrison, 1986).

The ratio of Br:Pb in the gasoline formulae can therefore be used only as an upper limit to predict the Br:Pb ratio in gasoline combustion aerosols transported to the ice core sites. Figure 8 shows a comparison between nsiBr and exPb measured in each ice core. Also illustrated is the upper limit of the amount of bromine expected from gasoline sources assuming the 2:1 Br:Pb ratio for aviation gasoline over the whole leaded gasoline era. World-wide leaded gasoline emissions were estimated to have peaked in 1970 C.E. (Thomas et al., 1997)—an assumption that is supported by the observed timing of the exPb maximum observed in both ice cores. Whilst it is likely that leaded fuel contributed to the increased bromine observed between 1925 and 1970, it is clear that it was not the only contributor to the nsiBr record, particularly after 1970 when the nsiBr record continues to rise despite a worldwide decline in leaded fuel consumption. The disparity between the exPb and nsiBr records suggests the driving force for the enhanced emission of Br was still active and increasing after 1970.

### 4.1.3 Seasonal salinity changes

Younger sea ice surfaces such as frost flowers, new and 1[st] year sea ice have a higher salinity and thus have higher bromine concentrations than older sea ice surfaces (Hunke et al., 2011) . The salinity of sea ice is at its maximum at the start of the winter season after which surface salinity slowly diminishes due to gravitational draining (Hunke et al., 2011). As summer approaches, ice continues to undergo desalination due to melting of surface snow which percolates through the ice (Hunke et al., 2011). Satellite observations that the BrO flux from the sea ice declines over summer (despite increasing insolation) is likely due to the combined reduction in young sea ice area and in ice salinity. Ocean surface salinity decreases in the summer due to the increased meteoric water flux and melting of desalinated sea ice. Salinity increases are therefore unlikely to be the sole cause of the nsiBr flux observed in the ice core records and the observed summer maximum in bromine.

### 4.1.4 Organic bromine species

Gaseous bromocarbons can be a source of inorganic bromine to the snow pack when they react with $\cdot OH$ or to a lesser extent with $\cdot NO_x$ or by photolysis (Kerkweg et al., 2008; WMO, 1995) to form the less reactive species $HBr$, $BrNO_3$ and $HOBr$. These species can then be washed out of the atmosphere and deposited on the snow surface due to their high solubility (Fan and Jacob, 1992; Sander et al., 1999; Yung et al., 1980).

The predominant source of gaseous bromine in the atmosphere is methyl bromide, $CH_3Br$ (WMO, 2002). The major modern sources of $CH_3Br$ are fumigation, biomass burning, leaded fuel combustion, coastal marshes, wetlands, rapeseed and the oceans (WMO, 2002). The ocean is also a major sink for $CH_3Br$, the temperature sensitive dissolution occurring through hydrolysis and chloride ion substitution to form bromide (WMO, 1995). ~30% of $CH_3Br$ was from industrial emissions at the time of the global peak in the $CH_3Br$ mixing ratio (1996-1998) (Montzka and Reimann, 2010). The timing of the massive increases in nsiBr seen at both ice cores sites coincides with the timing of maximum anthropogenic emissions of $CH_3Br$. However, the estimated 2.7 ppt increase in global tropospheric $CH_3Br$ above preindustrial levels equates to only ~ 3.7 ppt (0.05nM) Br incorporated into the snow pack (assuming 100% conversion efficiency of $CH_3Br$ in soluble Br species). This level is far less than the 2-5 nM increase in nsiBr observed in the ice cores during the industrial period.

Bromoform ($CHBr_3$) is emitted from vegetation such as marine phytoplankton and seaweed. It has the largest globe flux of all the bromocarbons (estimated at almost 5 times that of $CH_3Br$ (Kerkweg et al., 2008). However, it is very short-lived (atmospheric lifetime of ~ 17 days (Ordóñez et al., 2012) and

thus is confined to the marine boundary layer. Inorganic bromine formed from the destruction of $CHBr_3$ would therefore be representative of only local sources of organic bromine. The biological seasonal cycle maximises the production of $CHBr_3$ in summer and concentrations are greatly reduced but not negligible in winter (tidal forcing also influences bromocarbon emission by allowing coastal algae to dry-out (Kerkweg et al., 2008). The season of Arctic sea ice algae productivity is confined by limitations in available sunlight and nutrients resulting in a mid-to-late spring maxima – depending upon site location (Leu et al., 2015) – as is reflected in the seasonality of the MSA record. Direct transport of bromine enriched aerosols from these algal sources to the ice core sites again cannot explain the summer maximum of bromine observed in the ice. In addition to the incoherence of the seasonality of the bromine ice core signal, to-date biogenic sources have been considered insignificant sources of bromine in the Arctic marine boundary layer compared with the inorganic bromine source from sea salts (Simpson et al., 2007).

## 4.2    Cause of the spring-time increase in bromine flux

### 4.2.1    Bromine explosion events

Spring is the time of 'bromine explosion' events above sea ice. Sea salt aerosols passing through these BrO plumes can become enriched with bromine by adsorbing the gaseous species (Fan and Jacob, 1992; Langendörfer et al., 1999; Lehrer et al., 1997; Moldanová and Ljungström, 2001; Sander et al., 2003). Nghiem (2012) showed that these bromine rich air masses can then be elevated above the planetary boundary layer and transported hundreds of kilometres inland. Increasing the frequency and duration of the bromine explosion events would therefore likely increase the amount of bromine delivered to the ice core sites during spring without influencing the total aerosol flux and thus explain the shift in the bromine seasonal concentrations from a purely summer to a broad spring-summer maxima (Fig. 3).

Spring-time field studies at Ny Ålesund, Svalbard have shown positive correlation between atmospheric filterable bromine species and elevated levels of sulfate and nitrate (Langendörfer et al., 1999; Lehrer et al., 1997) suggesting that acidic, anthropogenic pollution may be the driver of the observed increases in annual bromine enrichment during the industrial period and seasonal shift.

### 4.2.2    Acidity effects on debromination

In remote, relatively clean environments such as the Arctic, even small increases in acidity are thought to affect the cycling of bromine in the snow pack (Finlayson-Pitts, 2003; Pratt et al., 2013; Sander et al.,

1999). In the laboratory, increasing the acidity of frozen (Abbatt et al., 2010) and liquid salt solutions
(Frinak and Abbatt, 2006; George and Anastasio, 2007) increased the yield of gas-phase $Br_2$ whilst at
the same time increasing the *solubility* of other bromine species, such as $HBr$. The uptake efficiency of
$HBr$ by acidic sulfate aerosols, for example, is estimated at 80% compared to 30% for sea salt aerosols
(Parrella et al., 2012). Interestingly, Abbatt (1995) demonstrated that $HBr$ is more than 100 times more
soluble in super-cooled sulfuric acid solutions than $HCl$. This may explain the cause of bromine
enrichment in the aerosol measured in the ice cores relative to the more abundant chlorine (Fig. S3).
The results of both the laboratory and field studies suggest that increasing snow/ice acidity in the Arctic
will likely enhance spring-time bromine explosion events above the sea ice whilst the increase in
solubility allows the termination products of the explosion to be transported away from the sites on the
surface of acidic aerosols. Increasing spring-time bromine aerosol concentrations would increase the
average annual bromine concentrations deposited on the ice sheet and could explain the nsiBr records
observed in both ice cores.
There are also significant periods over which the calculated nsiBr record shows negative values (e.g.
1815-1870 C.E. in Summit-2010 and 1860-1940 C.E. in Tunu). The negative values are a result of the
Total Br being less than that calculated by interpolation from the smoothed MSA record. Though the
sources of Br and MSA are linked – which is what provides the similarities between the general low
frequency trend of the two species, the atmospheric processing, transport and deposition of the two
species may be modified by different variables such as changes in atmospheric acidity, for example.
These variables cause the short term differences between the MSA and Total Br records preserved in
the ice so we believe it is not unreasonable to expect negative values in the calculated non-sea ice Br
record when the MSA and Total Br are close (essentially no nsiBr). The periods of negative nsiBr do
correspond to the timing of increased sulfate concentrations (due to volcanic or industrial activity) and
this could be an indication that the atmospheric sulfate concentrations do have some influence on the
production of either the MSA or Br records.
Figure 9 illustrates that of the two dominant acidic species preserved in the ice, $HNO_3$ (represented by
nitrate) shows the highest correlation to total bromine over sub-decadal time scales at both ice core sites.
Records were detrended with an 11 year running average before comparison to isolate the high
frequency components of each record. The bromine – sulfuric acid (represented by sulfate) correlation
is not significant. This is primarily because there is no bromine response to the dominant volcanic sulfate
spikes throughout the record. The large spikes in sulfate concentrations did not cause a depletion of
bromine preserved in the snowpack (Figure 9). This result might be expected if the increased acidity
caused more bromine to volatize. These results suggest that $HNO_3$ is the most influential of the MBL
acidic species in the processing and transport of Br on aerosols in the MBL.
**4.2.3 NOx and links to bromine**
The snow and atmospheric chemistries of bromine and nitrate ($NO_3^-$) are tightly linked. $NO_3^-$ is one of
the main sources of the •OH radical. The •OH radical can oxidize bromide salts and cause the release
of gas-phase bromine species (Abbatt et al., 2010; Chu and Anastasio, 2005; George and Anastasio,
2007; Jacobi et al., 2014). Morin et al. (2008) observed that the majority of nitrate that is deposited to
the snow surface is of the form $BrNO_3$ in coastal Arctic boundary layer. $BrNO_3$ forms by gas-phase
reaction of $BrO$ and $NO_2$. $BrNO_3$ is quickly adsorbed back onto the snow and aerosol surfaces due to
its high solubility. The heterogeneous hydrolysis of $BrNO_3$ to again release bromine species back into
the gas-phase has also been observed (Parrella et al., 2012) and can occur both during sunlight hours as
well as in the dark (Sander et al., 1999). However, the study of Thomas et al. (2012) into the cycling of
$NO_x$ and bromine species in the snowpack at Summit concluded that the presence of snow $NO_3^-$ would
suppress the emission of BrO from the snow pack and into the interstitial air.
In spring, when the greatest concentrations of $BrO$ are observed over the sea ice the atmospheric
concentrations of $NO_x$ species is rising. After 1900 C.E. there was, on average, a 60% increase in spring
$NO_3^-$ concentrations observed in Summit-2010 ice core (Fig. 3d) which, as discussed in Sect. 4.2.1, if
reflected in the concentration of acidic aerosols landing on the sea ice (specifically $HNO_3$
concentrations) would enhance the emission of $BrO$ into the MBL. Satellite imagery shows that bromine
in the form of $BrO$ is confined primarily to the atmosphere above sea ice (Schönhardt et al., 2012;
Wagner et al., 2001) but the presence of measurable bromine concentration hundreds of kilometres
inland preserved in the ice cores demonstrates that the bromine must be transported inland, just not in
the form of BrO. The reaction of atmospheric $NO_2$ with $BrO$ can produce the highly soluble $BrNO_3$
which will preserve the bromine in the aerosol allowing it to be transported inland. If there are high
$NO_3^-$ concentrations at the deposition site this will aid in fixing the bromine into the snow pack. This is
supported by the observation that $NO_3^-$ snow pack concentrations reach a maximum in summer,
coherent with bromine snow pack concentrations even though maximum Br emission from the sea ice
occurs in spring. So it appears that $NO_x$ in its different forms, as $NO_2$, $NO_3^-$, $HNO_3$, or $BrNO_3$ is
intertwined with Br as it cycles between the gas and condensed phases and as it is transported from sea
ice source to deposition site. Elevated levels of $NO_x$ over the Arctic could thus be the cause of the
deviation of the bromine record from the MSA, sea ice proxy record.
The high correlation between the preindustrial (1750-1850 C.E.) $NO_3^-$ and Br records (Fig. 9) supports

this observation of co-transport and sink of Br and $NO_3^-$ into the snow pack, though the natural sources of each are distinctly different. In the industrial era the low frequency temporal profile of the total bromine and nitrate records differ considerably, particularly at Summit (Fig. S15), apparently questioning the tight relationship observed before 1850. However, the positive correlation between the nitrate and the Br/MSA (Fig. 4) and nsiBr (Fig. 8) records is striking at both sites. The large relative increase in bromine (compared with MSA) during the era of high $NO_x$ pollution may point to a non-sea ice source of bromine linked to nitrate emissions or simply an increased spring-time emission and summer-time deposition of Br from sea ice sources.

Bromine and $NO_x$ species shared a common source in the 20[th] century through the combustion of leaded gasoline (Sect. 4.1.2). As discussed above, we observe that leaded fuel pollution reaching the Arctic began to decline after 1970 in-line with reduced global consumption, but the amount of bromine in-excess of natural sources (nsiBr) continued to increase – following the trends in $NO_x$ pollution (Fig. 8a). The continued increase in $NO_x$ despite the decline in leaded fuel combustion is attributed primarily to biomass burning, soil emissions and unleaded fossil fuel combustion (Lamarque et al., 2013). As the leaded fuel source of bromine began to decline, organic bromine pollutants continued to increase, as was discussed in Sect. 4.1.4. This can only account for a small fraction of the observed Br. The continued correlation between nitrate and nsiBr despite the decoupling of nitrate and bromine anthropogenic sources after 1970, suggests that nitrate pollution is likely influencing the processing of local, natural sources of bromine in the polar MBL, in effect increasing the mobility of the bromine and thus its flux and preservation in the ice sheet.

### 4.2.4  Consequences of nitrate driven increased bromine mobility in the Arctic

Plumes of BrO emitted from sea ice regions have been linked to mercury deposition events which lead to an increase in the bioavailability of toxic mercury species in polar waters (Parrella et al., 2012). Increased spring-time mobilization of bromine from the sea ice induced by anthropogenic nitrate could therefore increase the frequency and duration of these events and thus the mercury toxicity of the oceans. Increased atmospheric bromine concentrations would also increase the frequency of ozone depletion events (Simpson et al., 2007) thereby altering the oxidative chemistry of the polar MBL.

Whilst several studies have begun to explore bromine records from ice cores as a proxy for past sea ice conditions, the results of this study demonstrate that in an era of massive increases in atmospheric acidity the natural relationship between bromine and sea ice conditions can become distorted, precluding it from being an effective modern-day Arctic sea ice proxy.

## 5 Conclusion

In this study we have shown that high resolution MSA measurements preserved in ice cores can be used as a proxy for sea ice conditions (specifically the size of the marginal sea ice zone) along specific sections of the Greenland coast. The MSA records show that sea ice began to decline at the end of the LIA and again, more dramatically during the Industrial period. Also, unsurprisingly, the changes in sea ice conditions in the northern sites have been less dramatic than along the southern coastline. Comparison between the 260 year records of bromine and MSA presented in this study allow us to show that in the preindustrial era bromine concentrations preserved in the Greenland ice sheet are also likely linked to the local sea ice conditions. With the decline of sea ice in the modern era and the dramatic increase in acidic pollutants reaching the Arctic the sea ice-bromine connection is distorted, precluding it from being an effective, direct sea ice proxy during the industrial era. The introduction of $NOx$ pollution in particular, into the clean Arctic environment promotes mobilization of bromine from the sea ice, which in turn increases the bromine enrichment of the sea salt aerosols, forcing more bromine inland (particularly in spring) than would occur naturally. Nitrate has also been linked with the mechanism for preservation of bromine in the snowpack. The summer-time maximum of nitrate may therefore be responsible for the observed summer-time bromine maximum preserved in the ice cores. Whilst Northern Hemisphere pollution may prevent bromine from being an effective modern-day sea ice proxy in the Arctic, in Antarctica the anthropogenic flux of nitrate species is thought to be small in comparison with natural sources (Wolff, 2013), leaving room for the possibility that bromine may still be an effective proxy for local Antarctic sea ice conditions and for preindustrial sea ice reconstructions.

## Author contribution

Manuscript written and data analysis performed by O.J.M with expert editing by E.S.. Ice cores supplied by J.R.M.. Tunu ice core was collected and processed by O.J.M, J.R.M., N.J.C, M.S., R.H.R. under the leadership of Beth Bergeron. Ice cores dated by M.S., J.R.M.. ICP-MS and CFA measurements performed by O.J.M, J.R.M., N.J.C., L.L, D.P., M.S.. MSA measurements designed and performed by M.G., E.S.

## Acknowledgements

This research was funded by the National Science Foundation; grant numbers 1023672 and 1204176.

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

000

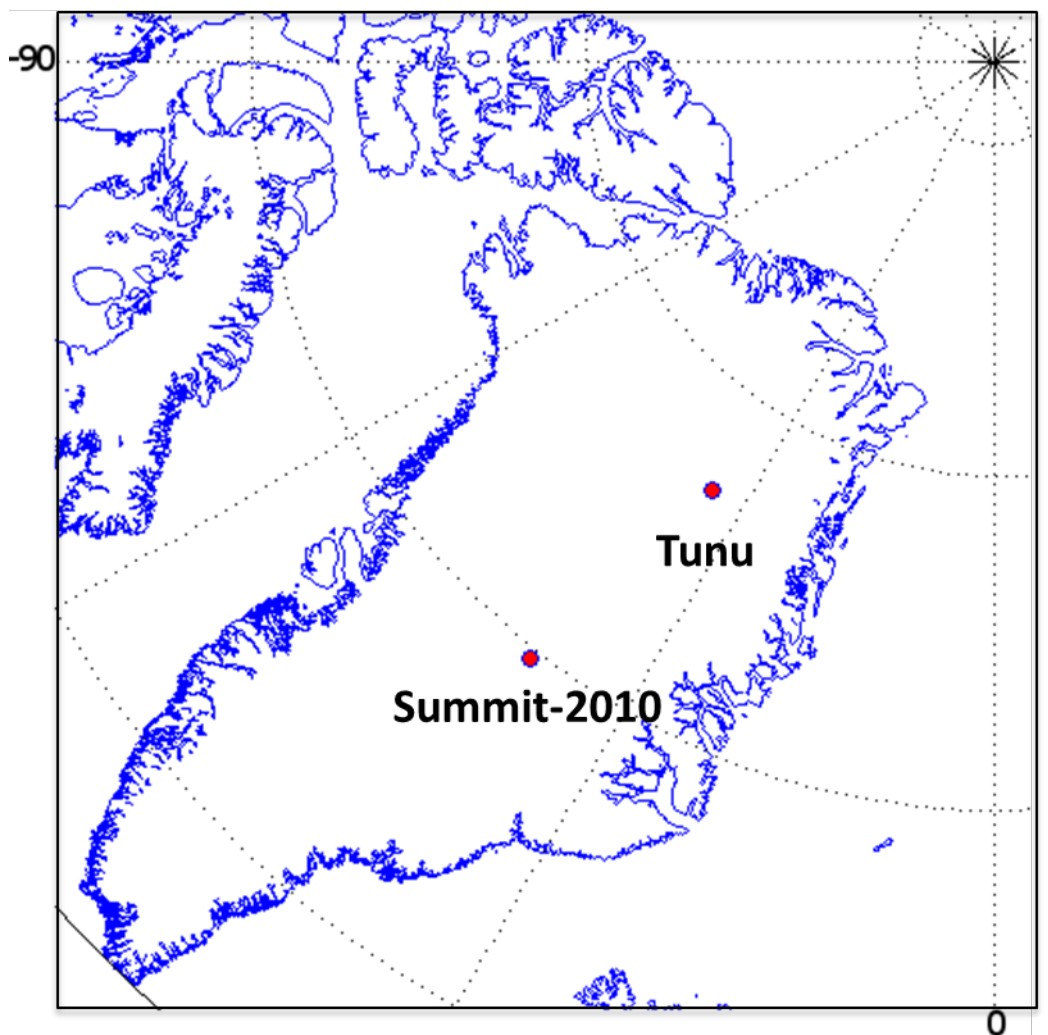

001

**Figure 1.** Locations of ice cores used in this study. Summit-2010: (72°20'N 38°17'24"W), Tunu: (78°
2' 5.5"N, 33° 52' 48"W)

004

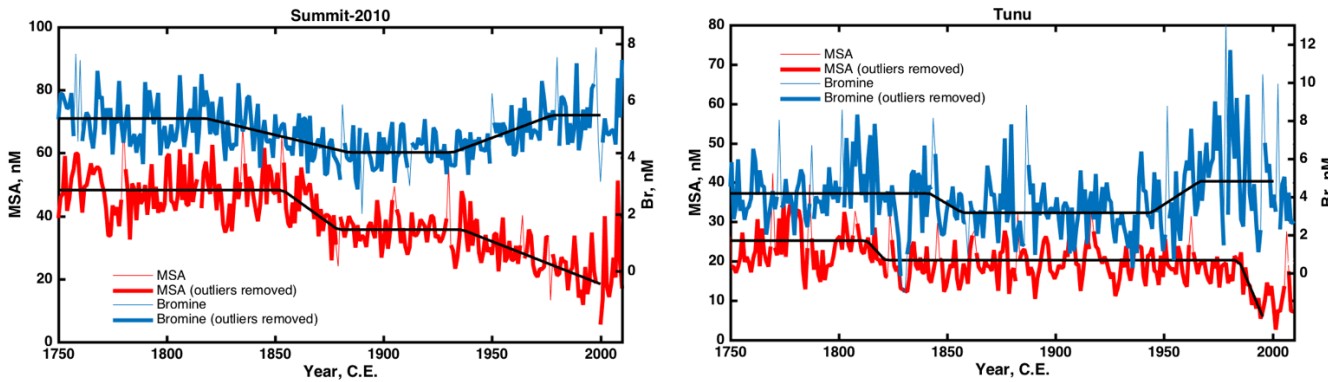

**Figure 2.** Annual record of bromine (thin blue) and MSA (thin red). Annual record of bromine (thick blue) and MSA (thick red) with outlying spikes removed using a 25 year running average filter described by Sigl et al. (2013). All records were fit with a 3 step linear regression (black) and the results of the fits which identify the timing of inflection points are summarized in Table S1. The time-series have been plotted to match the signal variability in the preindustrial era (1750-1850 C.E.).

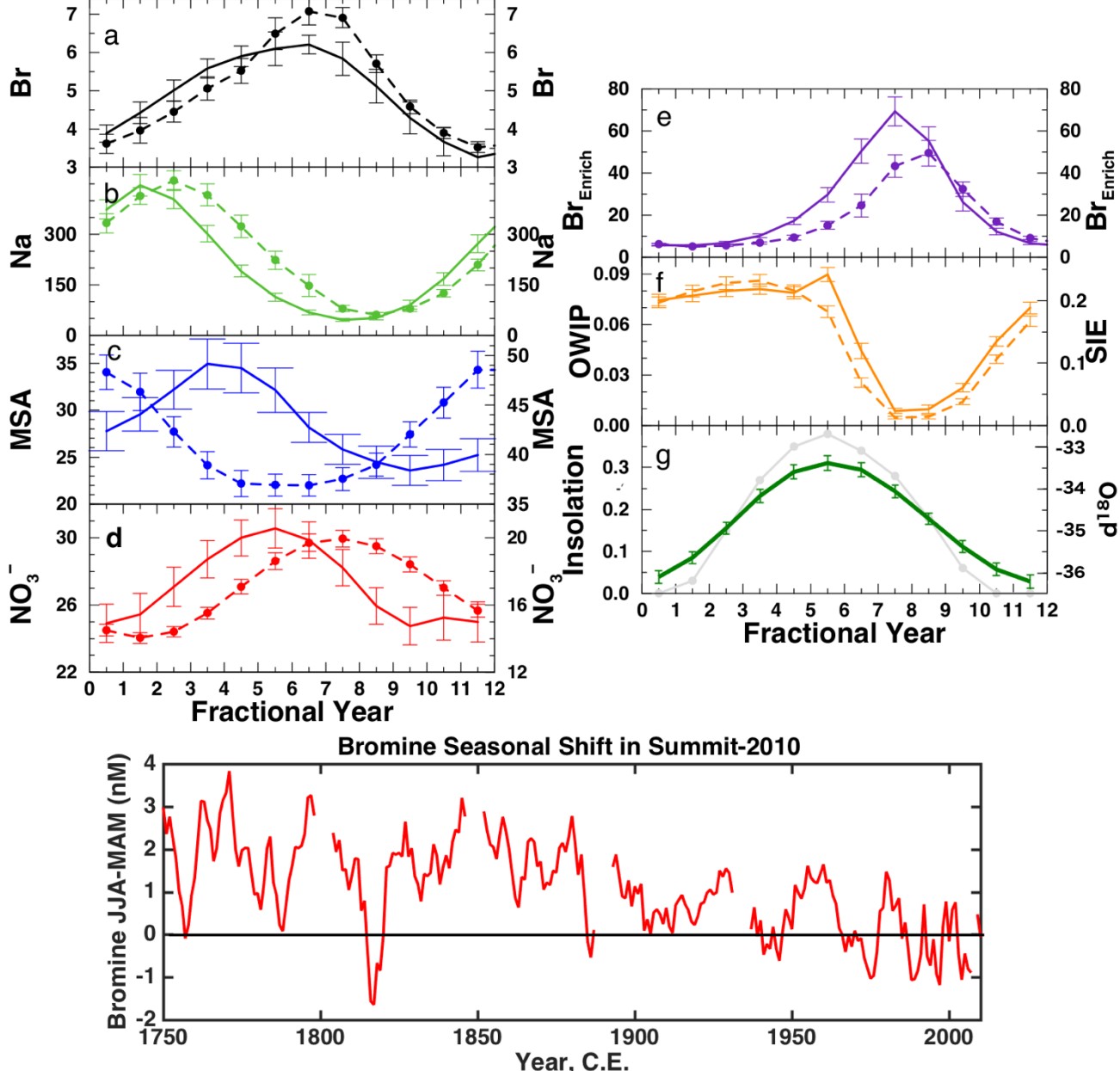

**Figure 3.** Upper plots: Average seasonal cycle of species in the Summit-2010 ice core. The left-hand Y axes are associated with the solid lines, and the right-hand Y axes associated with the dashed lines. Dashed lines (a-e): Average seasonal cycle from depths 43.5 – 87.3 m (years 1742-1900).  Solid lines (a-e): Average seasonal cycle from 0-43.5 m (years 1900-2010). Error bars indicate the standard error of the monthly value. (a) Total bromine, (b) total sodium, (c) MSA, (d) nitrate. Units for (a-d) are nM. Note that the seasonal cycle in bromine appears to broaden in the 1900-2010 period (see lower panel). Note also that the MSA maximum shifts from spring in the shallowest part of the ice core (solid line) to winter in the deepest part of the ice core (dashed line) due to post-depositional effects (see Fig. S1). (e)

Average seasonal cycle in bromine enrichment (relative to sea salt sodium, see Eq. (4)). (f-right) The sea ice extent (SIE, $x10^6$ km$^2$) within an area of the East Greenland coast [70°− 63° N, 15°− 45° W], (f – left) Area of open water within the sea ice pack (OWIP, $x10^6$ km$^2$) for the area defined by SIE. (g-left) Solar insolation at 12 GMT at the latitude of Summit (eosweb.larc.nasa.gov). (g-right) Annual cycle of the $\delta^{18}$O water signal averaged over 1900-2010 C.E.  Lower plot: Broadening of bromine seasonal cycle in the Summit-2010 ice core. The difference between the summer and spring bromine signal (JJA-MAM) was monitored over the length of the entire ice core. In the preindustrial era (pre-1850) bromine peaks in summer; realised as positive values of JJA-MAM. After 1900 there is a marked broadening of the seasonal signal towards spring and by ~1970 the seasonal signal maximum is routinely shared between summer and spring realised as an averaged JJA-MAM of approximately zero.

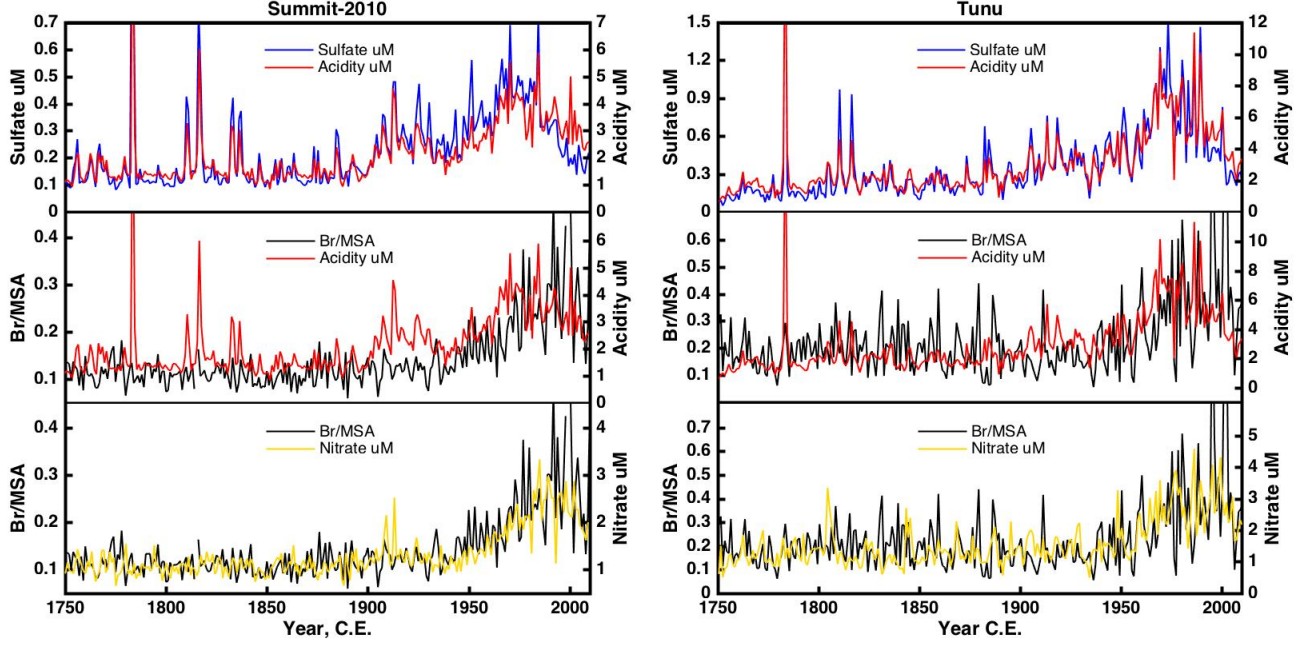

)34

)35

)36 **Figure 4.** Comparison between the measured total sulfur (shown as sulfate) and acidity records from
)37 each ice core (top panels). The acidity record is dominated by the influence of the sulfur species until
)38 the early 21[th] century when the $NO_x$ pollution remains elevated whilst anthropogenic sulfur sources are
)39 depleted resulting in a slight relative elevation of the total acidity relative to total sulfur concentrations.
)40 The large spikes in the acidity and sulfur records are identified as volcanic events. The ice core records
)41 cover the period of the 1783 Laki eruption as well as the Unknown 1909 eruption and Tambora eruption
)42 (Indonesia) in 1815 (Sigl et al., 2013). Comparison between Br/MSA and total acidity (center panels)
)43 and nitrate ($NO_3^-$, bottom panels) measured in the ice cores. The Br/MSA ratio follows the total acidity
)44 record closely except where the record is dominated by the sulfur component (e.g. early 1900s). Of the
)45 two major acidic species the Br/MSA follows the nitrate most closely at both ice core sites.

)46

)47

)48

)49

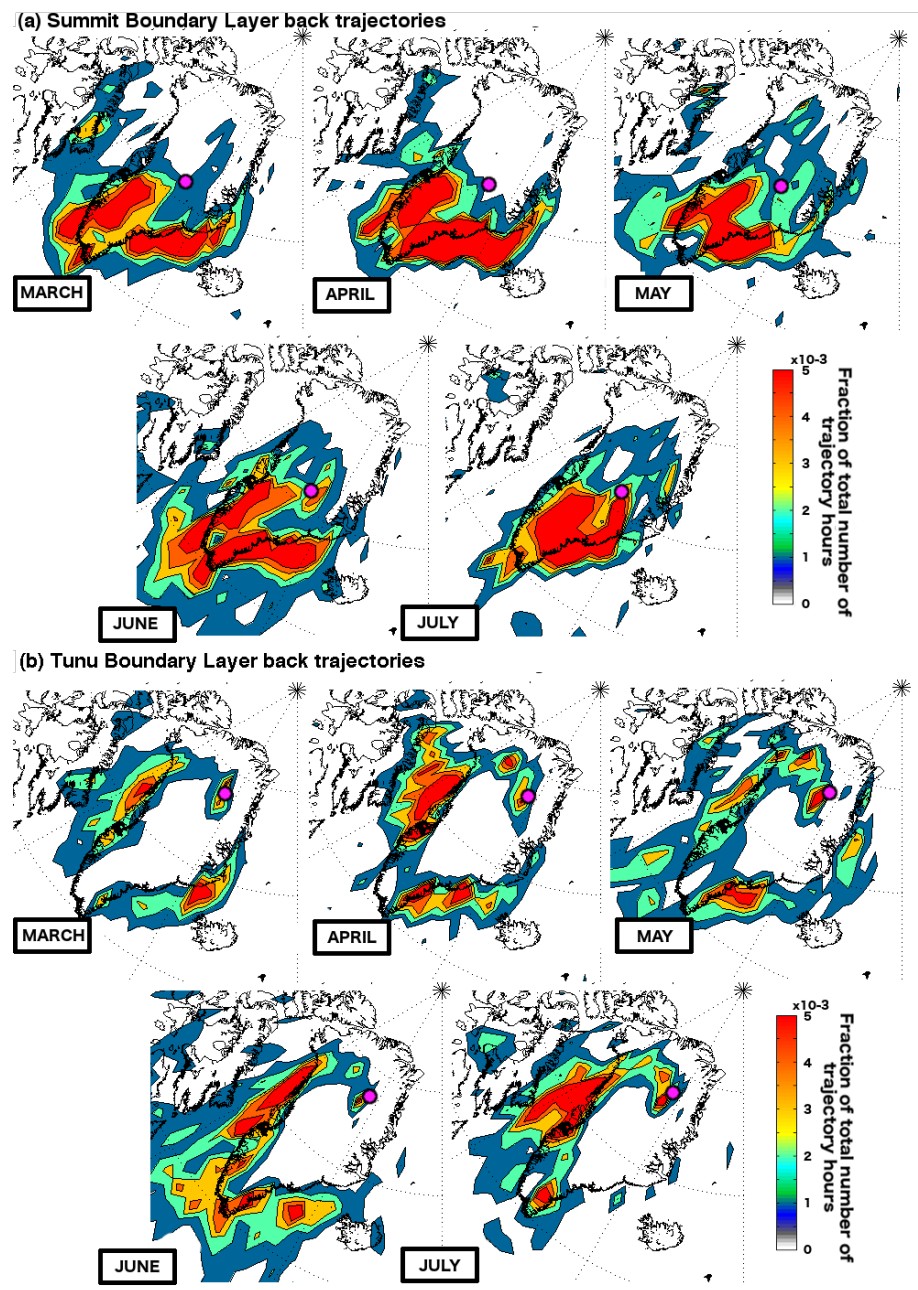

**Figure 5.** Air mass back trajectories from the (a) Summit-2010 and (b) Tunu ice core sites over the period 2005-2013 C.E. Maps display the fraction of the total number of trajectory hours (ranging between 21400-25500 hr month$^{-1}$) spent at altitudes under 500 m.  Back trajectories were allowed to travel for 10 days. New trajectories were started every 12 hours. Map grid resolution is 2°x 2°. Ice core locations are shown by a pink circle. Maps show that air masses consistently arrive at Summit from the SE Greenland coast with a smaller contribution from the SW coast. Air masses consistently arrive at Tunu from the western Greenland coast with a smaller contribution from the SE and NE coast. The air mass originating from the NE coast is most dominant in May and comparison with the total vertical column profile (Fig. S8) shows it is confined to lower altitudes unlike those from the west coast.

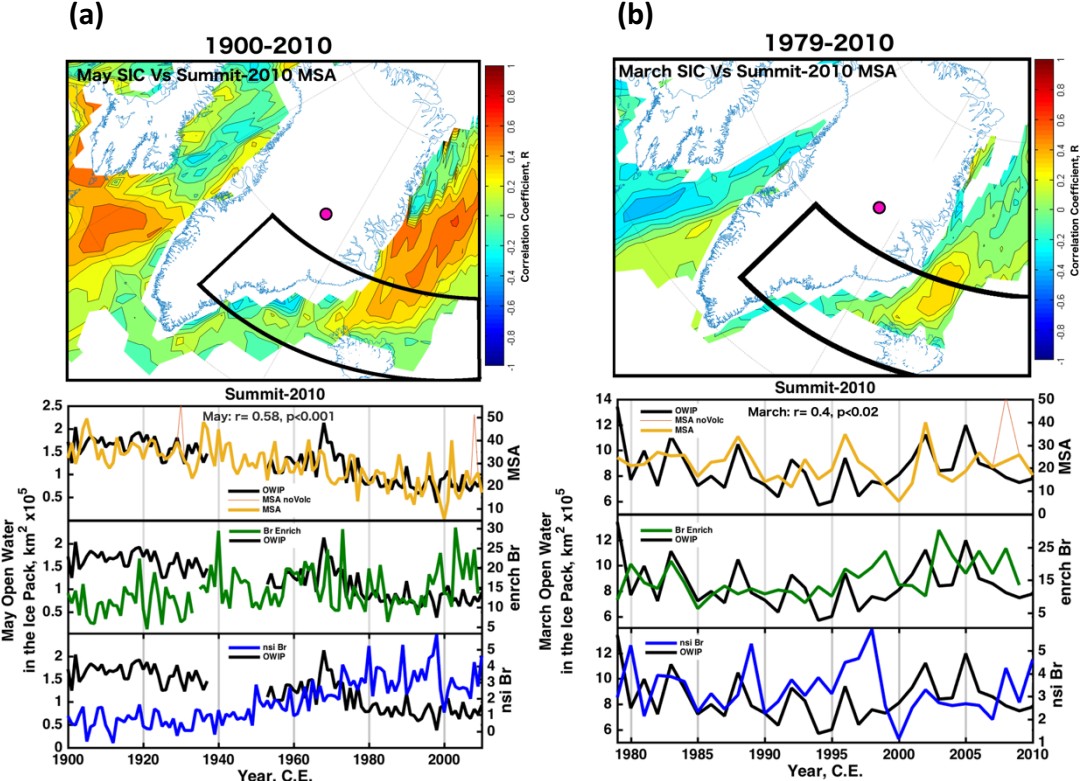

 **Figure 6.** Upper plot: Correlation map of monthly sea ice concentration (SIC) derived from the Summit-
 2010 ice core. The SIC map displayed corresponds to the month which shows the highest OWIP
 correlation (lower plot) with the annual MSA. Other monthly maps are shown in Fig. S9. (a) HadISST1
 ICE dataset from 1900-2010 C.E. correlated with annual records of MSA (with outlier removed). Only
 locations that showed a SIC variability greater than 10% and have a significant correlation (t-test,
 p<0.05) are displayed. The area of sea ice that is the likely source of MSA (as indicated by the air mass
 trajectories) are outlined in black [70°− 63°N, 0°− 45°W]. (b) As for (a) but focused on the satellite
 period 1979-2010 C.E. Lower plots: The correlation between the area of Open Water within the Ice
 Pack (OWIP) calculated within the black outlined areas shown on the upper maps and the annual MSA
 records (red, outliers removed − orange, nM). Summit-2010 MSA shows a significant, positive
 correlation with the amount of OWIP during spring within the integrated regions over both time periods.
 The highest correlations were found for March over the 1979-2010 period and May for the 1900-2010
 period. In (b) if the MSA source region is enlarged to [70°−63°N, 0°−60°W] the March OWIP/MSA
 correlation increases slightly (from 0.38 to 0.4). The Summit-2010 enrBr(Na) (nM) and nsiBr (nM)
 records are also compared to the same OWIP records. Particularly over the longer time period there is
 little correlation between the series.

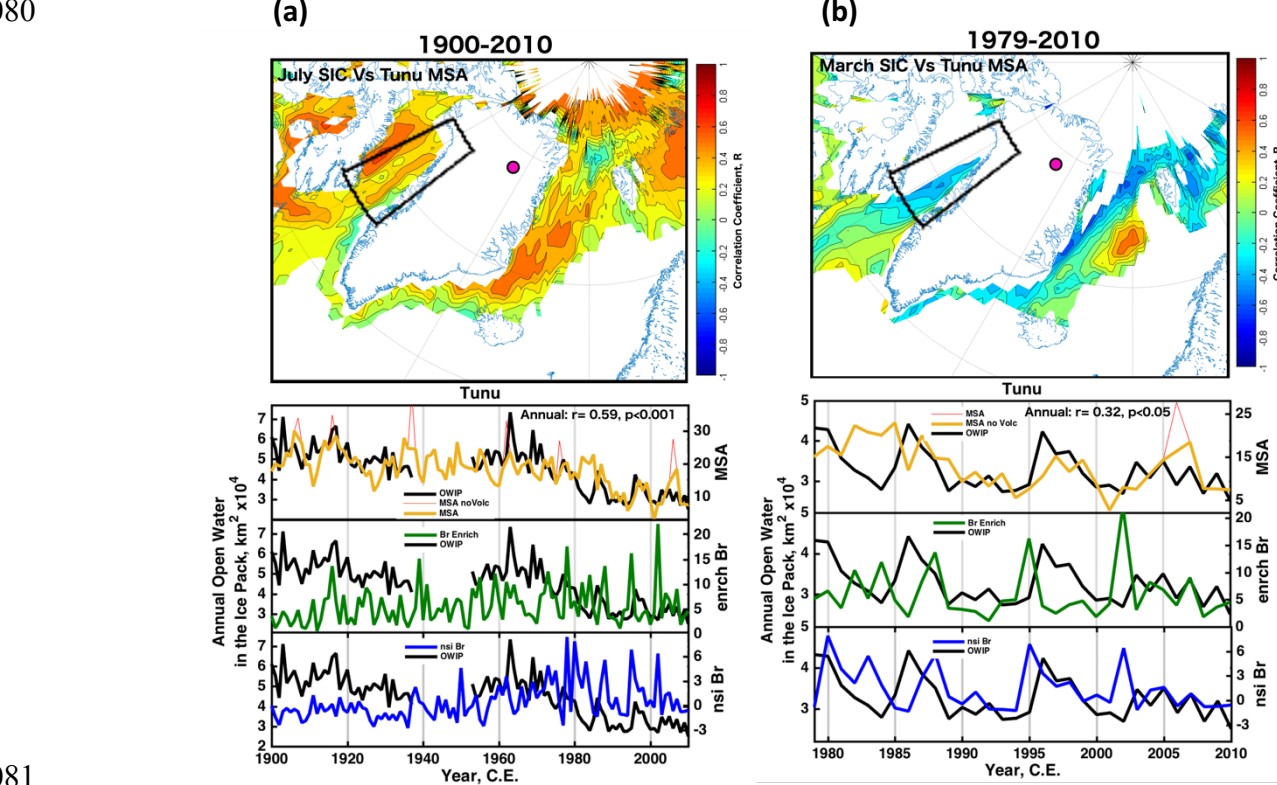

**Figure 7.** Upper plots: Correlation maps of monthly sea ice concentration (SIC) derived from the Tunu ice core. (a) HadISST1 ICE dataset from 1900-2012 C.E. correlated with annual records of MSA. The monthly SIC map displayed corresponds to the month which shows the highest OWIP correlation (lower plot) with the annual MSA. Other monthly maps are shown in Fig. S10. Only locations that showed a SIC variability greater than 10% and have a significant correlation (t-test, p<0.05) are displayed. The area of sea ice that is the likely source of MSA (as indicated by the air mass trajectories) are outlined in black [77°− 67°N, 62°−50°W]. (b) As for (a) but focused on the satellite period 1979-2012 C.E. Lower plots: The correlation between the area of Open Water within the Ice Pack (OWIP) calculated within the black outlined areas shown on the upper maps and the annual MSA records (red, outliers removed - orange). The Tunu enrBr(Na) (nM) and nsiBr (nM) records are also compared to the same OWIP records and show poor correlation, particulary over the longer time period.

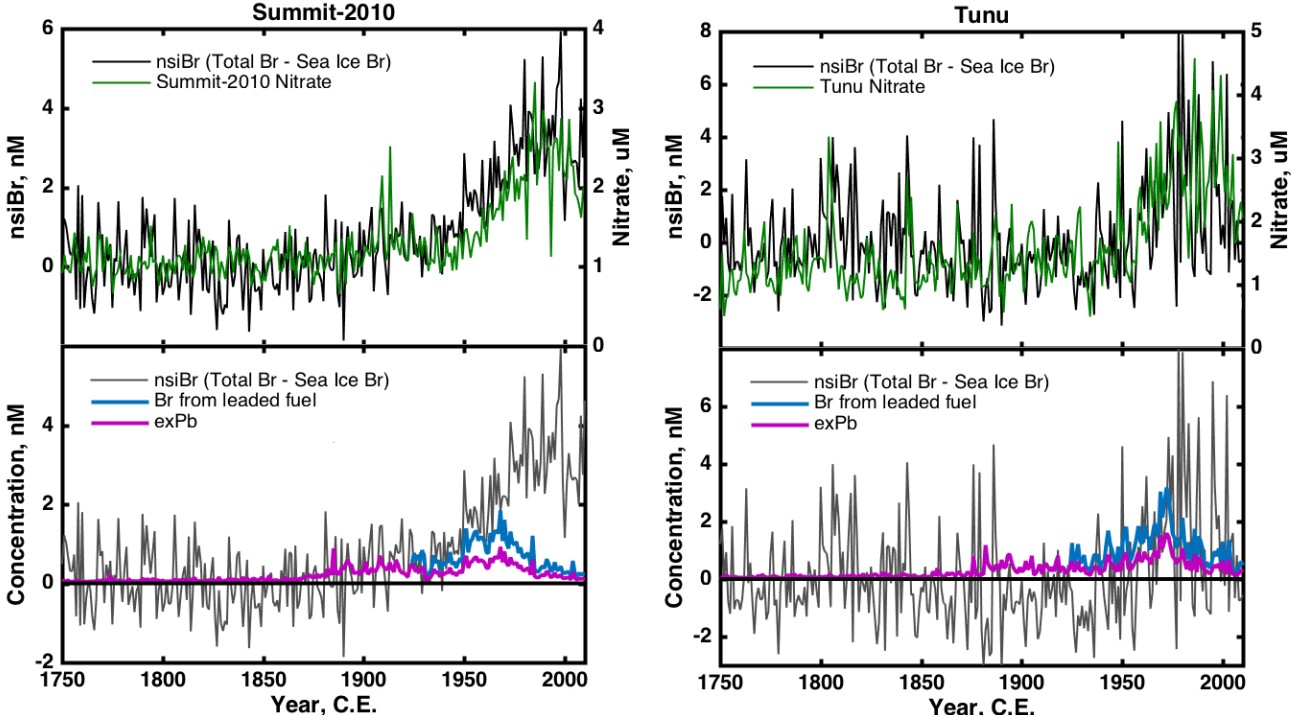

)96 **Figure 8.** Upper panels: Comparison between bromine in excess of what is expected from a purely sea

)97 ice source (nsiBr, black) and nitrate. The temporal similarities between the nitrate and nsiBr records are

)98 high and indicate that nitrate is a likely driving force for the enhanced release of bromine species from

)99 sea ice sources. Lower panels: Comparison between the calculated nsiBr record and excess lead (exPb,

purple) measured in the ice cores. The lower panels also show the upper limit to the amount of bromine
that could be derived from leaded fuel combustion by assuming exPb:Br ratio of 1:2 after 1925 (blue).
After 1970, when world consumption of leaded gasoline began to fall, nsiBr concentrations continued
to rise at both ice core sites far above the concentrations that could be explained by leaded gasoline
sources.



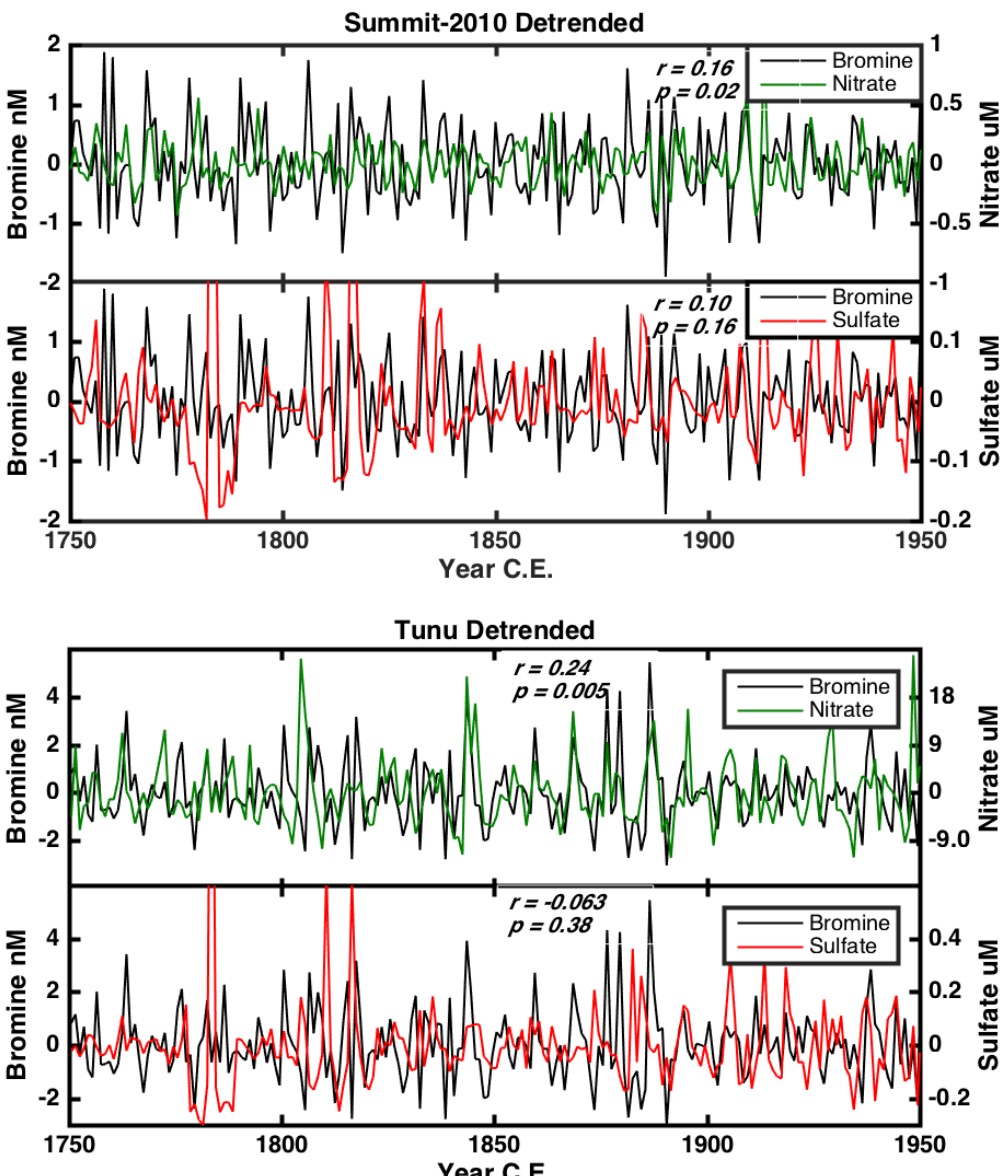

**Figure 9.** High frequency comparison between the annual bromine, nitrate and sulfate records measured in the ice cores. Each series has been detrended with an 11 year running average before comparison to remove the low frequency changes in each record. The correlation is highest between bromine and nitrate at both sites. The r-value for bromine versus nitrate at Summit increases in significance ($r = 0.24$, $p = 0.001$) when the entire period (1750-2010) is considered. At both sites there is a close relationship between the variability in the nitrate and bromine due to their intimate relationship during emission from the sea ice, transport and deposition onto the snow pack. The correlation between sulfate (or indeed bulk acidity) and bromine is not significant over any of the time periods shown at either site. Particularly

118    evident is the non-response of the bromine signal to the sulfur rich volcanic events as described in

119    Sect.4.2.2.

120