# Peer review of "Sea ice and pollution-modulated changes in Greenland ice core"

_Climate of the Past, 2016_

## Referee Comment (RC1) · Anonymous Referee #1 · 9 Jun 2016

General comments:

Present and past sea ice conditions play a decisive role assessing climate history and predicting future developments. Accordingly it is of pivotal importance to identify correspondingly reliable proxies. To this end this manuscript (ms) investigates the use of methansulfonic acid (MSA) and bromine records retrieved from two Greenland ice cores (Summit-2010 and Tunu) as a potential proxy for historic sea ice conditions in the Arctic. In addition pollution affected bromine deposition changes are considered. Both ice cores were analysed by current cutting-edge methods, followed by an elaborate and sound evaluation. The authors found that MSA can be primarily used as a proxy for the size of the marginal sea ice zone along specified regions of the Greenland coast line.

The same is true for bromine regarding the preindustrial era, while afterwards bromine records are significantly influenced by human activities. On the whole the ms at hand is a nice piece of work and an important contribution on this field of research. It is well written and structured and all parts, including figures, are essential. It clearly addresses the scientific scope of CP. I recommend a final publication after some (minor) revisions specified below.

Specific comments:

Chapter 2.1: Accurate absolute dating is pivotal for the subsequent correlation analyses. Please provide a reasonable error estimate for both ice cores and assess the potential impact on the correlations shown in Figs. 5-7.

Chapter 2.4, page 7, line 146-150 and Fig. 5: Did you use 10 day back trajectories (as stated in chapter 2.4) or 10 hr back trajectories as mentioned in Fig 5 (the latter seems unreasonable unless extremely high wind velocities prevailed) – please clarify.

Chapter 3.1, page 8, line 179-182 and Fig. 2 and Table S1: Please briefly describe the way you performed the "3 step linear regression" and how you identified the points of inflection.

Chapter 3.1, page 9, line 224-232 and Fig. S2: Albeit unusual, negative bromine enrichment relative to chlorine might as well be caused by a (positive) Cl enrichment relative to Na. Corresponding Cl vs. Na scatter plots could be instructive.

Chapter 4.2: While the increase of nss-related bromine (exBr) in the industrial era is scrutinised at length, I am missing an explanation for the late summer bromine maximum in the preindustrial era (although this point is insinuated in chapter 4.2.3 line 512-517). Note that this interesting finding is in contrast to the observed BrO concentration maximum in coastal Antarctic regions occurring mainly in spring (at Halley around October/November with an apparent small secondary maximum in March/April; Saiz-Lopez et al., Science, 317, 348-351, doi:10.1126/science.1141408, 2007). Surprisingly, however, in both Polar Regions bromine activation seems to roughly coincide with the respecting seasonal nitrate maximum (i.e. October/November for coastal Antarctica, see Wagenbach et al., J. Geophys. Res. 103(D9), 11007-11020, 1998). Do you think, a similar mechanism is valid in (still pristine) coastal Antarctica?

Chapter 4.2.3, page 18, line 518-519: To be honest, I cannot realize from these figures that nitrate and bromine records "differ dramatically" in the industrial era! An additional plot showing explicitly Br vs. nitrate could be enlightening.

Minor points: Page 4, lines 63-65: write Br-/Na+ or Br/Na (but not Br-/Na).

Page 9, line 226: Sander et al. (2003).

Page 11, line 296: It is actually Fig. 5b (and not Fig. 6b).

Page 15, line 400 and 407: The correct name is 1,2 dibromethane or 1,2 dibromethylen (i.e. BrH2C-CH2Br, abbreviated DBE) – 1,2 diethyl bromide nonexistent.

Page 25, line 725-728: Please refer to the respecting final paper (not the discussion paper): Sander, R., Keene, W. C., Pszenny, A. A. P., Arimoto, R., Ayers, G. P., Baboukas, E., Cainey, J. M., Crutzen, P. J., Duce, R. A., Hönninger, G., Huebert, B. J., Maenhaut, W., Mihalopoulos, N., Turekian, V. C., and Van Dingenen, R.: Inorganic bromine in the marine boundary layer: a critical review, Atmos. Chem. Phys., 3, 1301-1336, doi:10.5194/acp-3-1301-2003, 2003.

Table S1 (caption): inflection (not infection – witty typo!)

---

## Referee Comment (RC2) · Anonymous Referee #2 · 12 Jun 2016

Maselli and coauthors present a study investigating the behaviour of two possible sea ice proxies, Br and MSA, from two Greenlandic ice cores, Tunu and Summit. This is an interesting dataset and discussion, however I hold some significant doubts regarding the interpretations and methods. Most of my concerns relate to the production and interpretation of the MSA record. This would be the first time that a convincing MSA record has been published from Greenland ice cores, and the first time ever that such data has come from a continuous melting system. Given the significance of such data, it is surprising that the technique description and validation is so limited. I also have doubts regarding the assumption to calibrate Br trends using MSA, as the two have different production processes with respect to sea ice. Further, such a calibration is

performed for a period where there is no independent reliable observational data. Can the authors be sure they are not comparing proxy with different meaning? There is a lot of impressive data presented in this work but the authors should not put the cart before the horse: A thorough demonstration of the reliability of the data presented; a more complete investigation of the divergent trends between MSA and Br, and between Tunu and Summit; and a more detailed consideration of the potential impacts of acidity on the stability of Br and MSA in ice cores, would make this work a considerable contribution to the literature and our understanding of these important possible sea ice proxies. The following comments on the manuscript are divided into topics instead of referring to particular lines or sections.

Major comments MSA record Many papers have been written about the measurement of MSA (Legrand et al., 1993; Curran et al., 2001) as well as the problems of stability and mobility (Abram et al., 2008; Smith et al., 2004; Curran et al., 2002) as well as challenges of interpretation (Abram et al., 2013, Wolff et al., 2006; Legrand and Mayewski, 1995). Given the length of time that the community has required to tackle the distribution and stability of such a challenging ion, a thorough validation of the measurement method is required. Have replicate measurements been carried out? Have parallel discrete samples been measured (ie, samples that have not been exposed to the melter?). How do these results compare to other recent observations of MSA in Greenland (e.g Kuramoto et al., 2011, Jaffrezo et al., 1994)? Is it possible that MSA is destabilized at the melthead or when acid is added to the meltstream? These are critical questions as there is no pre-existing MSA record for comparison and in this manuscript MSA is used as a calibratiuon reference for the Br data.

Br measures Line 143. In the text the authors refer to Bromine as the sum of all bromine species that could be present into ice core samples. Equation 4 shows a calculation for Br enrichment, which has been used previously through the work of Spolaor and coauthors (2013) whereas in this manuscript the authors always use total bromine for discussion. Total Br follows sea salt and its components such as sodium. It is the

difference, or the additional amount of Br produced beyond the sea salt component, that must be evaluated, at least if you want to consider Br as a sea ice tracer.

Br summer peak and biological production Br emission from sea ice is well documented from both satellite observations of BrO and aerosol measurements. It is detected from the beginning of March, the arrival of solar radiation, to early June. The data show a maximum in Br in mid-June (also shown in figure S1). The results are in agreement with those obtained from NEEM (Spolaor et al., 2014) of maximum Br enrichment in summer. The authors claim that biological production could be a factor influencing the total Br concentration however I am not convinced of this for the following reasons: MSA, biologically produced, has its greatest snow concentration during spring, with consistently lower values during summer. Ardyna et al 2013 (Biogeosciences, 10, 4383–4404) report annual time series of surface chlorophyll for the Arctic and show that the main production is concentrated in spring with less production in summer. There could be an influence of biogenic bromine in spring but this is not supported by satellite observations, where the highest atmospheric concentrations are over sea ice. Release of biogenic bromine from sea ice through percolation is quite unlikely due to the low porosity of Arctic spring sea ice (Zhou et al 2013, JGR: Oceans, Vol. 118, 3172–3189, 2013) Finally, during summer satellites do not detect BrO in the polar atmosphere

MSA/Br correction (line 331) The authors suggest a correction for Br using MSA in the preindustrial era. To do this, a linear relationship has been used between the two parameters. I am not convinced about this approach because the relationship between Br, and its enrichment over seasonal sea ice is not linear but logarithmic. This is explained by Br chemistry over sea ice – very briefly, one Br radical produces two radicals causing the explosion. The relationship between MSA and the location of the sea ice edge, or as proposed by the authors, with open water leads in the ice pack, is linear. What is the correlation between Br and MSA? What is the relationship used? No chart has been presented in the paper although this is one of the central discussions of the

manuscript. In Fig 3 (and other work in the literature) it is shown that MSA can undergo remobilization (the seasonal cycle of MSA is changed completely) and smoothing in the ice. How much does this interfere with the Br correction? Again I mention my comment about the dangers of comparing a proxy with a proxy. Do MSA and Br represent the same source processes at the two sites investigated? Abram et al. (2013) have reviewed at least 5 different interpretations for MSA records around Antarctica alone. In this work, how can the authors correct a proxy (Br) influenced by the presence of seasonal sea ice with a proxy (MSA) that seems to quantify open water leads in the ice pack? Again my comment that more careful attention must be given to the discrepancies between the records. Infigure 2 there is a positive correlation until 1870 for Summit while for Tunu it persists until 1950. Then the correlation between Br and MSA seems negative. Why there is this discrepancy between the two cores? Tunu is a low accumulation site, Summit is far away from the coast; how much should these two sites be expected to demonstrate a similar sea ice signal? Are they even sampling the same sea ice regions? It has been demonstrated that sea ice cannot be reconstructed from MSA in central East Antarctica (Wolff et al., 2006) – a similar level of criticism needs to be applied to records from central Greenland. Br after 1950 The authors have investigated the Br increase of Br after 1950 in both cores. In the Summit core Br increases significantly from 1930 while in the Tunu core the increase is sharper and starts around 1950. The authors investigated different possible explanations, suggesting that anthropogenic pollution plays an extremely marginal role compared to biogenic contributions and acidity which can play a more central role. As discussed previously biological production can contribute to total bromine emission but a major influence on Br deposition is not supported by satellite observations of BrO. Acidity can play a role in the process, primarily for halogen recycling in sea salt aerosols and not so much for heterogeneous recycling on sea surfaces. Nevertheless, acidity is dependent on sea ice to have any effect: without sea ice the role of acidity would be zero. Therefore, the first order influence on Br recycling is from sea ice and a second or third order influence may be attributed to acidity. The links drawn between nitrate and Br are extremely

interesting but the data presented does not satisfactorily describe the influence of acidity. The authors are correct that the data points to more of an influence from nitrate than from sulphate, but none of the correlations in Fig 9 are strong, even if some are significant. How much of the significance between Br and nitrate can be ascribed to autocorrelation based on their seasonality?

Back trajectory and sea ice correlation Back trajectories shown in Figure 5 indicate that for Summit the main atmospheric pattern, or the source regions, originate from the south and south-east of Greenland while for Tunu the main source is Baffin Bay with sporadic incursions from southeast Greenland. Figure 6 shows correlation maps of monthly sea ice concentration (SIC) derived from the HadISST1 ICE dataset from 1900-2012. Why is it that the area used for correlation is different to the south and south-east secotrs of Greenland, where the Summit backtrajectories originate from? For panel b of figure 6, how is the correlation value in the graph (0.45) related to the data in the satellite maps, where none of the data approach a value of 0.45? For Tunu (Figure 7), as for Summit, the main correlation with sea ice is detected in northeast Greenland where BT analysis do not suggest an atmospheric source region. On what basis was the correlation window chosen? Again, why does the graph in panel b include a correlation of r=0.53 when this value is not shown in the correlation maps? For both sites, what would the correlations be if "outliers" were included? Other comments Line 101. Bromine is not normally measured in low resolution due to interference from 40Ar39K. What were the relative sizes of baseline to signal and was there a significant background when quantifying Br in low resolution? Line 225. Why has bromine been referenced to chloride? Chloride can undergo atmospheric reactions and is not recommended for evaluating Br enrichment. For consistency, as well as due to its stability, sodium should be used. Line 472. Although the effect of springtime Arctic haze is well described, the high springtime concentrations of sulphate and nitrate cited by the authors may be associated with biological production. Lehrer et al 1997 conducted their study in Ny-Alesund, a research facility right on the coast. Such a site is therefore susceptible to oceanic productivity, especially in connection to the springtime opening

of sea ice in the adjacent fjord. Line 557-559. Br and its enrichment in ice cores has been used for reconstructing sea ice changes in recent periods in the Arctic. The authors haven't provided a satisfying explanation for why bromine doesn't work at Tunu or Summit but does work at other Arctic sites. Acidity should play a role in influencing reactive halogens such as bromine but without sea ice no reactions will occur. It is further dissatisfying that the large acid peaks observed in the record do not have a consistent effect on either Br or MSA. Figure 2. It would be easier to look at the data if they do not overlap.

---

## Author Comment (AC1) · 3 Aug 2016

Chapter 2.1: Accurate absolute dating is pivotal for the subsequent correlation analyses. Please provide a reasonable error estimate for both ice cores and assess the potential impact on the correlations shown in Figs. 5-7. ——————————————
— The age error estimates for both ice cores have been added to the manuscript. The Tunu ice core was dated with an error of 1 year and Summit-2010 has an error estimate of 0.33 years. To assess the potential impact of the ice core dating errors on Figs. 6 and 7 (Fig. 5 does not involve the ice core records),We have re-plotted the maps using MSA time scales shifted to either extreme of the timescale error estimates (Figure R 1). The results of this analysis show that the timescale shifts do not change the areas of positive and negative correlation between sea ice concentration and the MSA records, but do change the magnitude of the correlation in some areas. I have only performed this analysis over the shorter, satellite time period (1979-2012) as this is where the effect would be seen most dramatically. The correlation for the longer record (1900-2012) is dominated by the large, low frequency changes in the MSA record and so is not dramatically affected by a temporal shift of up to 1 year. These plots have been added to the supplementary section of the manuscript and referenced in the text. –––––––––––––––––––––––––––––––––––––––––––––––––––––-

Chapter 2.4, page 7, line 146-150 and Fig. 5: Did you use 10 day back trajectories (as stated in chapter 2.4) or 10 hr back trajectories as mentioned in Fig 5 (the latter seems unreasonable unless extremely high wind velocities prevailed) – please clarify. –––––– ––––––––––––––––––––– 10 day back trajectories were used – as correctly described in Chapter 2.4 of the text. The figure captions of Fig 5 and S4 have both been corrected to also say that 10 day (not 10 hour) trajectories were used. –––––––––––––––––– ––––––––––––––––––––––––––––––––-

Chapter 3.1, page 8, line 179-182 and Fig. 2 and Table S1: Please briefly describe the way you performed the "3 step linear regression" and how you identified the points of inflection. The following has been added to Sect. 3.1 to explain how the '3 step linear regression' was performed: –––––––––––––––––––––––––– The "3 step linear regression" was performed by simultaneous linear least squares fitting of 3 straight lines joined by 'inflection points' to the data sets. The variables of the fitting procedure were the slopes and intercepts of each line as well as x-axis locations at which the total function switched from one linear section to the next (the inflection points). Initial guess values were supplied for each variable to help the fitting procedure reach reasonable values. –––––––––––––––––––––––––––––––––––––––––––––––––––––-

Chapter 3.1, page 9, line 224-232 and Fig. S2: Albeit unusual, negative bromine enrichment relative to chlorine might as well be caused by a (positive) Cl enrichment relative to Na. Corresponding Cl vs. Na scatter plots could be instructive. –––––––

—————————————— We thank the referee for this observation. Cl vs. Na scatter plots have been added to the supplementary figure and the following discussion included in the manuscript. "Both sites also show a (small) positive enrichment of chlorine relative to sodium, which is amplified at small sodium concentrations. Chlorine containing aerosols are expected to undergo similar chemical processing to bromine containing aerosols but the enrichment factors of bromine (relative to sodium) are much larger which is likely due to the high solubility of bromine species such as HBr (Sander et al., 2003). Alternatively, the chlorine enrichment could be interpreted as a sodium depletion of the aerosols particularly in those of small diameter where both concentrations are low; this would amplify the bromine enrichment (relative to sodium) but would not explain the bromine enrichment relative to chlorine. It is likely that both halogens undergo some degree of enrichment and the sodium undergoes some depletion in the aerosols, though it is difficult to determine this from the data." ——————————————

————————————————————————————————-

Chapter 4.2: While the increase of nss-related bromine (exBr) in the industrial era is scrutinised at length, I am missing an explanation for the late summer bromine maximum in the preindustrial era (although this point is insinuated in chapter 4.2.3 line 512-517). Note that this interesting finding is in contrast to the observed BrO concentration maximum in coastal Antarctic regions occurring mainly in spring (at Halley around October/November with an apparent small secondary maximum in March/April; Saiz-Lopez et al., Science, 317, 348-351, doi:10.1126/science.1141408, 2007). Surprisingly, however, in both Polar Regions bromine activation seems to roughly coincide with the respecting seasonal nitrate maximum (i.e. October/November for coastal Antarctica, see Wagenbach et al., J. Geophys. Res. 103(D9), 11007-11020, 1998). Do you think, a similar mechanism is valid in (still pristine) coastal Antarctica? ——— —————————————— While there may be a concentration threshold at which nitrate begins to significantly influence the bromine activation it is likely that the nitrate-bromine interaction mechanism is the same in pristine coastal Antarctica as it is in the preindustrial Arctic (which too was more 'pristine' than during the industrial period). We

have unpublished data from West coast Antarctic ice cores that show seasonal bromine maxima in Oct/Nov (with secondary peaks ∼June) which, as the Reviewer notes is co-incident with the seasonal nitrate maximum (Nov/Dec) and the satellite observations of BrO maxima in the neighboring sea ice. This may suggest that while the BrO emission from the sea ice is initiated by the increased spring insolation (as it also is in the Arctic), the deposition of the Br inland and its fixation into the snow is linked to its interaction with nitrate. This is supported by the work of Thomas et al. (2012) whose study of the cycling of NOx and bromine species in the snowpack at Summit concluded that the presence of snow nitrate would suppress the emission of BrO from the snow pack and into the interstitial air – in essence helping to preserve the bromine in the snow pack. More discussion highlighting these differences between Antarctic and Arctic records has been included in the manuscript to try and provide the explanation the reviewer requires regarding the summer maximum in Br. ————————————————————
————————————————————-

Chapter 4.2.3, page 18, line 518-519: To be honest, I cannot realize from these figures that nitrate and bromine records "differ dramatically" in the industrial era! An additional plot showing explicitly Br vs. nitrate could be enlightening. ————————————————
———— A plot comparing total Br and Nitrate at both sites has been included in the supplementary (Figure R 2) and referenced in the manuscript. The two time-series have been plotted to match the variability in the preindustrial era 1750-1850 C.E.. Hopefully this figure now supports the statement that Br and nitrate records are different – particularly at Summit. The difference is not as great at Tunu because the sea ice did not change as dramatically at Tunu as it did at Summit. ————————————————
————————————————————-

Minor points: Page 4, lines 63-65: write Br-/Na+ or Br/Na (but not Br-/Na). ————————
——————————————————— In the reference to which the discussion is referring the author (Spolaor et al., 2013) uses (Br-/Na) since the form of the bromine measured is as bromide ( it is isolated by ion chromatography) whilst the sodium is measured directly

by ICPMS so it is in its neutral form (Na). So the manuscript has not been changed as it is reflecting the discussion of Spolaor (2013). ———————————————————————————————————————————-

Page 9, line 226: Sander et al. (2003). The manuscript has been updated

Page 11, line 296: It is actually Fig. 5b (and not Fig. 6b). The manuscript has been updated ——————————————————————————————————————-

Page 15, line 400 and 407: The correct name is 1,2 dibromethane or 1,2 dibromethylen (i.e. BrH2C-CH2Br, abbreviated DBE) – 1,2 diethyl bromide nonexistent. ————————————————— This mistake has been corrected in the text at both locations. DEB changed to 1,2-dibromoethane (DBE). The cited references were also updated as they were not included in the reference list (Berg et al., 1983; Nriagu, 1990; Oudijk, 2010) ————————————————————————————————————-

Page 25, line 725-728: Please refer to the respecting final paper (not the discussion paper): Sander, R., Keene, W. C., Pszenny, A. A. P., Arimoto, R., Ayers, G. P., Baboukas, E., Cainey, J. M., Crutzen, P. J., Duce, R. A., Hönninger, G., Huebert, B. J., Maenhaut, W., Mihalopoulos, N., Turekian, V. C., and Van Dingenen, R.: Inorganic bromine in the marine boundary layer: a critical review, Atmos. Chem. Phys., 3, 1301-1336, doi:10.5194/acp-3-1301-2003, 2003. The manuscript has been updated ————————————————————————————————————————-

Table S1 (caption): inflection (not infection – witty typo!). The manuscript has been updated, thanks. ——————————————————————————————————————-

Figure Captions

Figure R 1: The effect of the timescale error on the correlation between ice core annual MSA concentrations and Sea ice concentration. The time scales for the MSA records at each ice core site were shifted to either extreme of the error in the time series dating and the correlation maps replotted. The effect of shifting the time series is to

change slightly the magnitude of the correlation at each location but not the sign of the correlation.

Figure R 2: Comparison between nitrate and bromine records at both ice core sites. The time-series have been plotted to match the signal variability in the preindustrial era (1750-1850 C.E.). The difference between the two time-series is most dramatic at the Summit-2010 site because the sea ice record changes most dramatically at this site also – and sea ice is the underlying driver of the bromine record.

References

Sander, R., Keene, W. C., Pszenny, A. A. P., Arimoto, R., Ayers, G. P., Baboukas, E., Cainey, J. M., Crutzen, P. J., Duce, R. A., Hönninger, G., Huebert, B. J., Maenhaut, W., Mihalopoulos, N., Turekian, V. C. and Van Dingenen, R.: Inorganic bromine in the marine boundary layer: a critical review, Atmos. Chem. Phys., 3, 1301–1336, doi:10.5194/acp-3-1301-2003, 2003. Spolaor, A., Vallelonga, P., Plane, J. M. C., Kehrwald, N., Gabrieli, J., Varin, C., Turetta, C., Cozzi, G., Kumar, R., Boutron, C. and Barbante, C.: Halogen species record Antarctic sea ice extent over glacial-interglacial periods, Atmos. Chem. Phys., 13, 6623–6635, doi:10.5194/acp-13-6623-2013, 2013. Thomas, J. L., Dibb, J. E., Huey, L. G., Liao, J., Tanner, D., Lefer, B., von Glasow, R. and Stutz, J.: Modeling chemistry in and above snow at Summit, Greenland – Part 2: Impact of snowpack chemistry on the oxidation capacity of the boundary layer, Atmos. Chem. Phys., 12(14), 6537–6554, doi:10.5194/acp-12-6537-2012, 2012.

Sea Ice Concentration correlated with Summit-2010 MSA

| MARCH | APRIL | MAY | JUNE | JULY |
|---|---|---|---|---|

**Summit-2010 MSA timescale shifted 4 months earlier**

**Summit-2010 MSA timescale not shifted**

**Summit-2010 MSA timescale shifted 4 months later**

Sea Ice Concentration correlated with Tunu MSA

| MARCH | APRIL | MAY | JUNE | JULY |
|---|---|---|---|---|

**Tunu MSA timescale shifted 1 year earlier**

**Tunu MSA timescale not shifted**

**Tunu MSA timescale shifted 1 year later**

**Fig. 1.**

[Figure]

[Figure]

**Fig. 2.**

---

## Author Comment (AC2) · 3 Aug 2016

Maselli and coauthors present a study investigating the behavior of two possible sea ice proxies, Br and MSA, from two Greenlandic ice cores, Tunu and Summit. This is an interesting dataset and discussion, however I hold some significant doubts regarding the interpretations and methods. Most of my concerns relate to the production and interpretation of the MSA record. This would be the first time that a convincing MSA record has been published from Greenland ice cores, and the first time ever that such data has come from a continuous melting system. Given the significance of such data, it is surprising that the technique description and validation is so limited. I also have doubts regarding the assumption to calibrate Br trends using MSA, as the two have

different production processes with respect to sea ice. Further, such a calibration is performed for a period where there is no independent reliable observational data. Can the authors be sure they are not comparing proxy with different meaning? There is a lot of impressive data presented in this work but the authors should not put the cart before the horse: A thorough demonstration of the reliability of the data presented; a more complete investigation of the divergent trends between MSA and Br, and between Tunu and Summit; and a more detailed consideration of the potential impacts of acidity on the stability of Br and MSA in ice cores, would make this work a considerable contribution to the literature and our understanding of these important possible sea ice proxies. The following comments on the manuscript are divided into topics instead of referring to particular lines or sections. Major comments MSA record Many papers have been written about the measurement of MSA (Legrand et al., 1993; Curran et al., 2001) as well as the problems of stability and mobility (Abram et al., 2008; Smith et al., 2004; Curran et al., 2002) as well as challenges of interpretation (Abram et al., 2013, Wolff et al., 2006; Legrand and Mayewski, 1995). Given the length of time that the community has required to tackle the distribution and stability of such a challenging ion, a thorough validation of the measurement method is required. Have replicate measurements been carried out? Have parallel discrete samples been measured (ie, samples that have not been exposed to the melter?). How do these results compare to other recent observations of MSA in Greenland (e.g Kuramoto et al., 2011, Jaffrezo et al., 1994)? Is it possible that MSA is destabilized at the melthead or when acid is added to the meltstream? These are critical questions as there is no pre-existing MSA record for comparison and in this manuscript MSA is used as a calibratiuon reference for the Br data.

Several updates to the manuscript have been added in order to address the concerns of the referee regarding the reporting of the MSA measurement technique and its valida-tion – since it is the first time the ESI/MS/MS analytical technique has been combined with a continuous, online sampling of an ice core. The following three plots have been added to the supplementary section of the paper and referenced in the manuscript.

A plot comparing the MSA record from this study and that from Legrand 1997 (Legrand et al., 1997) - both collected near Summit (as mentioned in the text) has been added to the Supplementary material and referenced in the manuscript ( see Figure R 1). The Legrand record is from the GRIP ice core drilled at the location 7234'N, 3738'W in 1993, 35 km NE of the Summit-2010 ice core site. The MSA samples from the GRIP core were isolated in the field, 1-2 hours after the ice cores were retrieved and subsequently processed via discrete sample ion chromatography. In comparison, the Summit-2010 ice core was measured using the new technique of continuous melting of the ice core combined with continuous analysis by electrospray triple-quad mass spectrometry (ES/MS/MS - as described in the manuscript). As was mentioned in the text, the analytical technique of ES/MS/MS for discrete MSA ice core samples is well established; studies using this technique were published by the Saltzman group over 10 years ago (Saltzman et al., 2006) so the new part of the technique is using the ES/MS/MS in continuous mode.

The tight overlap between the low frequency trend of the two MSA series demonstrates that the new, continuous measurement technique is able to achieve a comparable accuracy in MSA concentration measurements to the discrete technique. It also demonstrates that negligible amounts of MSA are being lost during the continuous melt method (by destabilization at the melt-head as the review suggests). No acid is added to the melt stream that feeds the MSA into the ES/MS/MS so MSA would not be degraded or destabilized in this way either- as the reviewer suggests. The reviewer also questions the validity of using the Summit-2010 MSA record as a calibration for the Br record since there "is no pre-existing MSA record for comparison". We believe that the Summit-2010 MSA replicates the 'pre-existing' Legrand GRIP MSA record (which is also from the Summit region). It was for this reason that the Summit-2010 MSA record was not presented as unique record of MSA from Summit, instead it was used so that an accurate comparison between MSA and Br could be drawn since they were measured simultaneously from the same ice core.

To demonstrate the reproducibility of the MSA measurements and thus the robustness of the online, continuous technique, several plots have been included in the supplementary section which show replicate MSA measurements from the same depth of the Tunu ice core (Figure R 2). Each section of the ice core was cut into multiple longitudinal sticks so that replicate measurements of all chemical species could be performed at any depth. The same stick in each core section was melted and analyzed to give the primary record of all the chemical species measured online. Replicate measurements were then performed using the ice sticks which 'overlapped' the depths of interest. The online, continuous MSA technique reproduces both the high and low frequency components of the MSA record extremely well in the duplicate analyses even though the duplicate measurements were often performed days or even months apart. The records demonstrate both the high precision and temporal stability of the technique. Replication at such high resolution also demonstrates that the high frequency component of the MSA is actually real variability of MSA preserved in the ice core and not just the noise of the analytical technique. The entire MSA record from the Tunu ice core was also reproduced by discrete analysis of MSA samples collected from the CFA system. The discrete samples were collected as the continuous measurements were performed by directing part of the sample stream into an autosampler collection system just before they entered the continuous analyzer. The samples were then frozen and later measured using ion chromatographic separation and the ESI/MS/MS detection. In this plot (Figure R 3 and added to the supplementary section) the continuous data have been averaged over the same depth range covered by each discrete sample and then both series plotted as the average age over that depth range. Over the 1750-2012 C.E. period the Tunu discrete measurements were, on average, 7% higher than the online measurements (dashed lines indicate average values over the 1750-2012 period). Both the discrete and continuous samples experienced identical conditions from ice melt to collection so the reason for the offset in measured concentration is likely due to differences in post-processing of the data (such as calibration) and not due to loss of MSA in the continuous, online technique.

—————————————-

Br measures Line 143. In the text the authors refer to Bromine as the sum of all bromine species that could be present into ice core samples. Equation 4 shows a calculation for Br enrichment, which has been used previously through the work of Spolaor and coauthors (2013) whereas in this manuscript the authors always use total bromine for discussion. Total Br follows sea salt and its components such as sodium. It is the difference, or the additional amount of Br produced beyond the sea salt component, that must be evaluated, at least if you want to consider Br as a sea ice tracer. ———
———————————————

We do not agree that it is the bromine (Br) in excess of the sodium concentration (exBr(Na)) or Br enrichment relative to sodium (enrBr(Na)) that must be used to consider Br as a sea ice tracer. Unfortunately, the use of sodium to try and extract the pure sea water component of the Br is complicated by the fact that a lot of sodium comes from the sea ice surface as well as from the open ocean. Sodium itself has been used as a sea ice proxy in several prominent studies (Wais Divide Project Memebers, 2013; Wolff et al., 2003) because, like Br, sodium is incorporated into the brine encapsulated within the sea ice which can then soak into the snow that sits on the surface of the sea ice and subsequently be blown aloft to produce the atmospheric sodium signal seen in the ice core. In addition, the sodium concentration is fractionated upon the formation of the ice when mirabolite ($Na_2SO_4$) is precipitated out of the brine solution at -8C (Abbatt et al., 2012). In saying this, the records of Total bromine and enrBr(Na), for example, do look similar except for two dips in the enrBr(Na) at years ∼1954 and 1990 where the total sodium signal has maxima and the low frequency variability is not as great in the enrBr(Na) record (Figure R 4). As bromine is a relatively new proxy for sea ice, this plot has now been included in the supplementary section so it can aid comparison to the work of Spolaor (2013a, 2014, 2016) and this discussion added to Sect. 4 of the manuscript. The reviewer states that "Total Br follows sea salt and its components such as sodium". This study has found that while there are similarities between the

total bromine and total sodium records it is not clear that they follow each other. Below is a plot (Figure R 5) comparing the total bromine and total sodium records from the Summit-2010 ice core. To avoid confusion between the bromine enrichment as defined by Spolaor and coauthors (2013b) or Brexc defined by Spolaor (2016) and exBr described in this study, the label of exBr has been changed to nsiBr (non sea ice Bromine) within the manuscript. ——————————————————————————————
———————————-

Br summer peak and biological production Br emission from sea ice is well documented from both satellite observations of BrO and aerosol measurements. It is detected from the beginning of March, the arrival of solar radiation, to early June. The data show a maximum in Br in mid-June (also shown in figure S1). The results are in agreement with those obtained from NEEM (Spolaor et al., 2014) of maximum Br enrichment in summer. The authors claim that biological production could be a factor influencing the total Br concentration however I am not convinced of this for the following reasons: MSA, biologically produced, has its greatest snow concentration during spring, with consistently lower values during summer. Ardyna et al 2013 (Biogeosciences, 10, 4383–4404) report annual time series of surface chlorophyll for the Arctic and show that the main production is concentrated in spring with less production in summer. There could be an influence of biogenic bromine in spring but this is not supported by satellite observations, where the highest atmospheric concentrations are over sea ice. Release of biogenic bromine from sea ice through percolation is quite unlikely due to the low porosity of Arctic spring sea ice (Zhou et al 2013, JGR: Oceans, Vol. 118, 3172–3189, 2013) Finally, during summer satellites do not detect BrO in the polar atmosphere. ——————————————————————

We agree that biogenic sources of bromine are likely insignificant in the ice core records. The text of Sect. 4.1.4 has been reworded to try and better reflect this. ——————————————————————————————————————-

MSA/Br correction (line 331) The authors suggest a correction for Br using MSA in

the preindustrial era. To do this, a linear relationship has been used between the two parameters. I am not convinced about this approach because the relationship between Br, and its enrichment over seasonal sea ice is not linear but logarithmic. This is explained by Br chemistry over sea ice – very briefly, one Br radical produces two radicals causing the explosion. The relationship between MSA and the location of the sea ice edge, or as proposed by the authors, with open water leads in the ice pack, is linear. What is the correlation between Br and MSA? What is the relationship used? No chart has been presented in the paper although this is one of the central discussions of the manuscript. ————————————————————

In the reviewer's explanation of the bromine explosion event they say that "one Br radical produces two radicals causing the explosion". We think that they mean one Br2 molecule produces two radicals as this is the mechanism as we understand it. The Br2 is produced when one non-reactive atmospheric Br species is adsorbed onto the ice surface (HOBr) which, when it reacts with the Br salt in the ice (BrÂŕ) releases Br2 and subsequently two reactive atmospheric Br radicals are produced (Frieß, 2004). The mechanism is labelled as a 'bromine explosion' because there is an exponential increase in reactive atmospheric bromine species. Also, it is important to note that this bromine release mechanism only occurs in acidic ice substrates (Frieß, 2004). It is important to note that in this study total bromine is measured – which is not isolated to just the reactive atmospheric species released in the bromine explosion events, so it wouldn't be expected that there is a exponential (or logarithmic as suggested by the reviewer) relationship between the atmospheric Br species measured in this study and the Br species in the sea ice. In fact it is primarily the soluble, non reactive bromine species which would be able to be transported the large distances to the ice core sites on the surfaces of aerosols and thus preserved in the ice core. Supplementary figures have been added which describe the process used to 'correct' the total bromine record using the MSA (Figure R 6). The figures include the correlation between MSA and Br which was used to determine the sea ice component of the bromine signal. From the Summit-2010 correlation it is easiest to see that the relationship between Br and

[Figure]

MSA does indeed appear to be closer to linear than either logarithmic or exponential. Also, the fitting function that was used to fit the MSA time series (a 264 point stienman function in the original manuscript) has been updated to a 9th order polynomial fit to simplify the analysis - the final record of the non-sea ice component of the bromine (exBr or nsiBr) is unchanged by this update. ————————————————————————— —————————————————-

In Fig 3 (and other work in the literature) it is shown that MSA can undergo remobilization (the seasonal cycle of MSA is changed completely) and smoothing in the ice. How much does this interfere with the Br correction? Again I mention my comment about the dangers of comparing a proxy with a proxy. ————————————————————

In Figs. 3 and S3 we demonstrate that both the bromine and MSA signal show a seasonal shift along the length of the ice core. The shift is on a sub-annual scale for both species so that the annual average value for each species appears to be unchanged. For this reason, the comparison between MSA and bromine was performed at an annual, not sub-annual resolution so the remobilization of the species along the length of the ice cores should not significantly effect this comparison. Although comparing proxy with proxy may be 'dangerous' it is common practice: such as the comparison between sodium and bromine that the reviewer suggests as a more appropriate way to develop the sea ice proxy. ————————————————————————————————————-

Do MSA and Br represent the same source processes at the two sites investigated? Abram et al. (2013) have reviewed at least 5 different interpretations for MSA records around Antarctica alone. In this work, how can the authors correct a proxy (Br) influenced by the presence of seasonal sea ice with a proxy (MSA) that seems to quantify open water leads in the ice pack? ————————————————————————

Areas of open water in the ice pack (OWIP) become relaminated with fresh sea ice (unless they represent a polynya) and it is the newest sea ice which is most saline and thus the major source of bromine enriched aerosol. In this case the size of the OWIP

influences the amount of fresh sea ice that is formed in these zones. In this way the two source processes are linked but are not identical. ——————————————————————————————————————————————-

Again my comment that more careful attention must be given to the discrepancies between the records. In figure 2 there is a positive correlation until 1870 for Summit while for Tunu it persists until 1950. Then the correlation between Br and MSA seems negative. Why there is this discrepancy between the two cores? Tunu is a low accumulation site, Summit is far away from the coast; how much should these two sites be expected to demonstrate a similar sea ice signal? Are they even sampling the same sea ice regions? ——————————————————————

We are not suggesting that the two sites have the same sea ice signal. Indeed, while there are similarities between the Br and MSA records at the two sites, they are distinctly different (as was demonstrated when the time series were fitted with the 3-step linear regression and the fitting parameters summarized in Tables S1and S2). As was shown by the air mass back trajectories the sea ice source regions for each ice core are on different sides of Greenland and at different latitudes so it is understandable that the sea ice signals are different – as discussed in the text. ——————————————————————————————————————————————-

It has been demonstrated that sea ice cannot be reconstructed from MSA in central East Antarctica (Wolff et al., 2006) – a similar level of criticism needs to be applied to records from central Greenland ——————————————————————

In their 2006 manuscript, Wolff (Wolff et al., 2006) stated that MSA from a central East Antarctic site (Dome C) could not be used as a sea ice proxy since MSA can be lost to the atmosphere post-deposition at very low accumulation sites. The EPICA Dome C (EDC) ice core used in the Wolff study has an average accumulation rate of 25 kg m-1 yr-1(Augustin et al., 2004) which is much lower than the accumulation rates of both Greenland ice cores used in this study. Indeed Wolff references the work of

Weller (2004) who was able develop a relationship between %MSA(and nitrate) loss
and average snow accumulation for a series of Antarctic ice cores. Extrapolation of this
relationship predicts 0% MSA (and nitrate) loss at accumulation rates of >105 kg.m-
1.yr-1. The average accumulation rates for ice cores used in this study were: Summit-
2010; 0.22 m yr-1 (220 kg m-1 yr-1) and 0.11 m yr-1 (110 kg m-1 yr-1) which are both
larger than the minimum accumulation rate needed to ensure 0% post-depositional
loss of MSA ( and nitrate) according to the work of Weller (2004). For this reason post-
depositional loss of MSA has not been included in this manuscript. However, comment
as to why the post-deposition loss of MSA is not considered has now been included
in the manuscript (Sect 3.2) ————————————————————————————————
————-

Br after 1950: The authors have investigated the Br increase of Br after 1950 in both
cores. In the Summit core Br increases significantly from 1930 while in the Tunu
core the increase is sharper and starts around 1950. The authors investigated dif-
ferent possible explanations, suggest- ing that anthropogenic pollution plays an ex-
tremely marginal role compared to biogenic contributions and acidity which can play a
more central role. As discussed previously biological production can contribute to total
bromine emission but a major influence on Br deposition is not supported by satellite
observations of BrO. Acidity can play a role in the process, primarily for halogen recy-
cling in sea salt aerosols and not so much for heterogeneous recycling on sea surfaces.
Nevertheless, acidity is dependent on sea ice to have any effect: without sea ice the
role of acidity would be zero. Therefore, the first order influence on Br recycling is from
sea ice and a second or third order influence may be attributed to acidity. The links
drawn between nitrate and Br are extremely interesting but the data presented does
not satisfactorily describe the influence of acidity. The authors are correct that the data
points to more of an influence from nitrate than from sulphate, but none of the corre-
lations in Fig 9 are strong, even if some are significant. How much of the significance
between Br and nitrate can be ascribed to autocorrelation based on their seasonality?
————————————————————————

The types and locations of the sources of nitrate and bromine and distinctly different, so it is unlikely that the correlation between the magnitudes of the low frequency components of the two species is dominated by autocorrelation based solely on their seasonality (See Figure 8a). The reviewer also questions whether the correlations determined in Fig. 9 (between only the high frequency components of the acidic species and the bromine) may be dominated by the the fact the two species show the same seasonality. However, the correlations were performed on annual averages of the bromine and nitrate so the seasonal characteristics of each species are essentially removed before the correlations were performed. The reviewer also suggests that biological production of bromine is not a major influence on deposited Br since it is not supported by satellite BrO observation. Satellite observations of this reactive form of atmospheric Br show that BrO is confined to areas directly above sea ice - minimal BrO is observed inland. As the bromine records generated in this study were from inland ice core sites this shows that bromine is transported inland – but must not be in the form of the species that are observed by the satellites – BrO. So, as the reviewer noted, "biological production can contribute to total bromine emission" and biological production is related to sea ice coverage and the amount of open water in the ice pack (OWIP), which is what is promoted in this study. Indeed more discussion has been added reference to the influence of nitrate in fixating the bromine into the snowpack in summer as the likely cause of the summertime bromine maximum observed in the ice core (more details included in the reply to reviewer 1) The manuscript describes the influence of nitrate in promoting more bromine to be released from the sea ice than would be in the absence of the anthropogenic species in addition to helping to fix the bromine in the snow pack. To aid in this discussion we have incorporated a plot into Figure 8 (Figure R 7) showing a direct comparison between the nitrate records and the nsiBr (non sea ice bromine) in addition to expanding the discussion of the links between nitrate and bromine in the emission, transport and deposition of bromine (Sect. 4.2.3) —————————————— ————————————————————————-

Back trajectory and sea ice correlation Back trajectories shown in Figure 5 indicate

that for Summit the main atmospheric pattern, or the source regions, originate from the south and south-east of Greenland while for Tunu the main source is Baffin Bay with sporadic incursions from southeast Greenland. Figure 6 shows correlation maps of monthly sea ice concentration (SIC) derived from the HadISST1 ICE dataset from 1900-2012. Why is it that the area used for correlation is different to the south and south-east secotrs of Greenland, where the Summit back-trajectories originate from? ————————————————

We thank the reviewer for this observation – indeed the secondary source region (as mentioned in the text) was highlighted for both ice core sites in figure 6 and 7 - this was a mistake and has been corrected in the manuscript. Now the primary source regions (as determined from the back trajectories from Fig. 5) are outlined in black on figures 6 and 7 and the lower, OWIP plots have been updated to display the OWIP values for the indicated source regions. The figure captions have been updated to try and make this more clear and more detail has been added to the analysis of the correlations between the MSA and OWIP records. In addition, the resolution of the maps has been increased to aid visualization. ————————————————————————————————
-

For panel b of figure 6, how is the correlation value in the graph (0.45) related to the data in the satellite maps, where none of the data approach a value of 0.45? For Tunu (Figure 7), as for Summit, the main correlation with sea ice is detected in northeast Greenland where BT analysis do not suggest an atmospheric source region. On what basis was the correlation window chosen? Again, why does the graph in panel b include a correlation of r=0.53 when this value is not shown in the correlation maps? For both sites, what would the correlations be if "outliers" were included? ——————————
————————————

The correlation values displayed in the lower, line plots are for the correlation between the area of Open Water in the Ice Pack (OWIP) within the black squares (the back-trajectories source regions) and the MSA time series. The upper, correlation maps

are displaying the direct correlation between the Sea Ice Concentration values (SIC) at each latitude and longitude displayed and the MSA time series. So these two parts of the figure are distinctly different, that is why the correlation values in the maps and the line plots are not related. ——————————————————————————————————————————-

Line 101. Bromine is not normally measured in low resolution due to interference from 40Ar39K. What were the relative sizes of baseline to signal and was there a significant background when quantifying Br in low resolution? ——————————————————

The bromine was measured at medium resolution - this resulted in sufficient mass separation between Br 79 and 40Ar39K to avoid interference between the two species. This has been corrected in the manuscript. ———————————————————————————————————————-

Line 225. Why has bromine been referenced to chloride? Chloride can undergo atmospheric reactions and is not recommended for evaluating Br enrichment. For consistency, as well as due to its stability, sodium should be used. ——————————————————

The manuscript has been updated to include the bromine enrichment relative to sodium as well as chlorine. ——————————————————————————————————————-

Line 472. Although the effect of springtime Arctic haze is well described, the high springtime concentrations of sulphate and nitrate cited by the authors may be associated with biological production. Lehrer et al 1997 conducted their study in Ny-Alesund, a research facility right on the coast. Such a site is therefore susceptible to oceanic productivity, especially in connection to the springtime opening of sea ice in the adjacent fjord. ——————————————————

Discussion about the cause of the seasonal nitrate cycle has now been added to the manuscript. "Although there are biological sources of nitrate in the ice core aerosol

source regions, in a recent study focused on the and record in the Summit-2010 ice core, Chellman et al. (2016) concluded that the preindustrial (1790-1812 C.E.) seasonal cycle was driven by biomass burning emissions. However, in the modern era (1930-2002 C.E.) oil-burning emissions became the dominant source of in the snowpack. The change in the dominant source due to industrialisation is the cause of the shift in timing of the seasonal cycle." ——————————————————————————
———————————-

Line 557-559. Br and its enrichment in ice cores has been used for reconstructing sea ice changes in recent periods in the Arctic. The authors haven't provided a satisfying explanation for why bromine doesn't work at Tunu or Summit but does work at other Arctic sites. Acidity should play a role in influencing reactive halogens such as bromine but without sea ice no reactions will occur. It is further dissatisfying that the large acid peaks observed in the record do not have a consistent effect on either Br or MSA. ————————————————————————

Figure R 4 has been added to aid direct comparison between the previous studies that used Br enrichment and this study which used total bromine. Also more discussion has been included which hopefully helps to explain how the large amounts of anthropogenic nitrate will effect the deposition of the bromine in the snow and thus distort the low frequency component of the bromine record from what it would be if it was purely a sea ice proxy. The fact that the large acidity peaks do not appear to have a consistent effect on the bromine records supports our conclusion that it is not total acidity (nitric acid + sulphuric acid + other minor acids) which effects bromine activation on the snow/ice but more specifically just HNO3. ——————————————————————————
——————————-

Figure 2. It would be easier to look at the data if they do not overlap. ——————————
————————————

Figure 2 has been replotted (Figure R 8) so that the data does not overlap. The timeseries are plotted so that the magnitude of the preindustrial variability is the same at each site. ————————————————————————————————————————————————————

FIGURE CAPTIONS

Figure R 1. Comparison between the MSA record from the the GRIP ice core drilled in 1993 (Legrand (1997)) and the Summit-2010 ice core drilled in 2010 (this study). Figure R 2. Demonstration of the reproducibility of the MSA online, continuous measurements performed on the Tunu ice core. Two different depths of the Tunu ice core are shown where the replicate analysis was performed by melting a secondary stick of ice cut from the same ice core and overlapping in depth: (a) Six 'overlap' ice cores were melted sequentially to replicate the MSA record over the depth 8-14 m.(b) Two 'overlap' ice cores were melted sequentially over the depth 186.2-187.9 m. Zooming in on a small section of the record at each depth demonstrates that the high frequency signal is real (not noise) and well replicated by the continuous MSA technique. Figure R 3: Comparison between discrete and continuous, online measurements of MSA measurements from the Tunu ice core. Figure R 4. Total bromine and bromine enrichment (relative to sodium) from the Summit-2010 ice core. The time-series have been plotted to match the signal variability in the preindustrial era (1750-1850 C.E.). Figure R 5. Total bromine and total sodium records from the Summit-2010 ice core. The concentrations of sodium were measured as, on average, 60 times greater than that of bromine and the temporal trends of both species is also quite different. The time-series have been plotted to match the signal variability in the preindustrial era (1750-1850 C.E.). Figure R 6: Summary of the technique used to determine nsiBr: the amount of Bromine in excess of what is expected from a purely sea ice source.(a,b) Blue dots, blue fit line: correlation plots between total bromine and total MSA in Summit-2010 and Tunu ice cores, respectively over the preindustrial period 1750-1880 C.E.. Red dots, yellow fit line: Correlation plots between total bromine and smoothed MSA time series shown in c and d.(c,d) Annual MSA record fit with 9th order polynomial. (e,f) Comparison between the total bromine record (black) and the bromine predicted from the smoothed MSA,

Br linear relationship determined in a and b (blue) – the bromine from a purely sea ice source. The difference between the blue and black lines is the amount of bromine in excess of what is expected from a purely sea ice source (nsiBr; See Fig. 8). Figure R 7: Upper panels: Comparison between bromine in excess of what is expected from a purely sea ice source (nsiBr, black) and nitrate. The temporal similarities between the nitrate and nsiBr records are high and indicate that nitrate is a likely driving force for the enhanced release of bromine species from sea ice sources. Lower panels: Comparison between the calculated nsiBr record and excess lead (exPb, purple) measured in the ice cores. The lower panels also show the upper limit to the amount of bromine that could be derived from leaded fuel combustion by assuming exPb:Br ratio of 1:2 after 1925 (blue). After 1970, when world consumption of leaded gasoline began to fall, nsiBr concentrations continued to rise at both ice core sites far above the concentrations that could be explained by leaded gasoline sources. Figure R 8: Annual record of bromine (thin blue) and MSA (thin red). Annual record of bromine (thick blue) and MSA (thick red) with outlying spikes removed using a 25 year running average filter described by Sigl et al. (2013). All records were fit with a 3 step linear regression and the results of the fits which identify the timing of inflection points are summarized in Table S1.The time-series have been plotted to match the signal variability in the preindustrial era (1750-1850 C.E.).

REFERENCES

Abbatt, J. P. D., Thomas, J. L., Abrahamsson, K., Boxe, C., Granfors, A., Jones, A. E., King, M. D., Saiz-Lopez, A., Shepson, P. B., Sodeau, J., Toohey, D. W., Toubin, C., von Glasow, R., Wren, S. N. and Yang, X.: Halogen activation via interactions with environmental ice and snow in the polar lower troposphere and other regions, Atmos. Chem. Phys., 12(14), 6237–6271, doi:10.5194/acp-12-6237-2012, 2012. Augustin, L., Barbante, C., Barnes, P. R. F., Barnola, J. M., Bigler, M., Castellano, E., Cattani, O., Chappellaz, J., Dahl-Jensen, D., Delmonte, B., Dreyfus, G., Durand, G., Falourd, S., Fischer, H., Flückiger, J., Hansson, M. E., Huybrechts, P., Jugie, G., Johnsen, S.

J., Jouzel, J., Kaufmann, P., Kipfstuhl, J., Lambert, F., Lipenkov, V. Y., Littot, G. C., Longinelli, A., Lorrain, R., Maggi, V., Masson-Delmotte, V., Miller, H., Mulvaney, R., Oerlemans, J., Oerter, H., Orombelli, G., Parrenin, F., Peel, D. a, Petit, J.-R., Raynaud, D., Ritz, C., Ruth, U., Schwander, J., Siegenthaler, U., Souchez, R., Stauffer, B., Steffensen, J. P., Stenni, B., Stocker, T. F., Tabacco, I. E., Udisti, R., Van De Wal, R. S. W., Van Den Broeke, M., Weiss, J., Wilhelms, F., Winther, J.-G., Wolff, E. W., Zucchelli, M., EPICACommunityMembers, Augustin, L., Barbante, C., Barnes, P. R. F., Barnola, J. M., Bigler, M., Castellano, E., Cattani, O., Chappellaz, J., Dahl-Jensen, D., Delmonte, B., Dreyfus, G., Durand, G., Falourd, S., Fischer, H., Flückiger, J., Hansson, M. E., Huybrechts, P., Jugie, G., Johnsen, S. J., Jouzel, J., Kaufmann, P., Kipfstuhl, J., Lambert, F., Lipenkov, V. Y., Littot, G. C., Longinelli, A., Lorrain, R., Maggi, V., Masson-Delmotte, V., Miller, H., Mulvaney, R., Oerlemans, J., Oerter, H., Orombelli, G., Parrenin, F., Peel, D. a, Petit, J.-R., Raynaud, D., Ritz, C., Ruth, U., Schwander, J., Siegenthaler, U., et al.: Eight glacial cycles from an Antarctic ice core., Nature, 429(6992), 623–628, doi:10.1038/nature02599, 2004. Chellman, N. J., Hastings, M. G. and McConnell, J. R.: Increased nitrate and decreased $\delta$15N–NO3$-$ in the Greenland Arctic after 1940 attributed to North American oil burning, Cryosph. Discuss., 1–22, doi:10.5194/tc-2016-163, 2016. Frieß, U.: Dynamics and chemistry of tropospheric bromine explosion events in the Antarctic coastal region, J. Geophys. Res., 109, 1–15, doi:10.1029/2003JD004133, 2004. Legrand, M., Hammer, C., De Angelis, M., Savarino, J., Delmas, R., Clausen, H. and Johnsen, S. J.: Sulfur-containing species (methanesulfonate and SO4 ) over the last climatic cycle in the Greenland Ice Core Project (central Greenland) ice core, J. Geophys. Res., 102(C12), 26663, doi:10.1029/97JC01436, 1997. Saltzman, E. S., Dioumaeva, I. and Finley, B. D.: Glacial/interglacial variations in methanesulfonate (MSA) in the Siple Dome ice core, West Antarctica, Geophys. Res. Lett., 33(11), 1–4, doi:10.1029/2005GL025629, 2006. Spolaor, A., Vallelonga, P., Plane, J. M. C., Kehrwald, N., Gabrieli, J., Varin, C., Turetta, C., Cozzi, G., Kumar, R., Boutron, C. and Barbante, C.: Halogen species record Antarctic sea ice extent over glacial-interglacial periods, Atmos. Chem. Phys., 13,

6623–6635, doi:10.5194/acp-13-6623-2013, 2013a. Spolaor, A., Gabrieli, J., Martma, T., Kohler, J., Björkman, M. B., Isaksson, E., Varin, C., Vallelonga, P., Plane, J. M. C. and Barbante, C.: Sea ice dynamics influence halogen deposition to Svalbard, Cryosph., 7(5), 1645–1658, doi:10.5194/tc-7-1645-2013, 2013b. Spolaor, A., Vallelonga, P., Gabrieli, J., Martma, T., Björkman, M. P., Isaksson, E., Cozzi, G., Turetta, C., Kjær, H. A., Curran, M. A. J., Moy, A. D., Schönhardt, A., Blechschmidt, A.-M., Burrows, J. P., Plane, J. M. C. and Barbante, C.: Seasonality of halogen deposition in polar snow and ice, Atmos. Chem. Phys., 14(18), 9613–9622, doi:10.5194/acp-14-9613-2014, 2014. Spolaor, A., Opel, T., McConnell, J. R., Maselli, O. J., Spreen, G., Varin, C., Kirchgeorg, T., Fritzsche, D., Saiz-Lopez, A. and Vallelonga, P.: Halogen-based reconstruction of Russian Arctic sea ice area from the Akademii Nauk ice core (Severnaya Zemlya), Cryosph., 10, 245–256, doi:10.5194/tcd-9-4407-2015, 2016. Wais Divide Project Memebers: Onset of deglacial warming in West Antarctica driven by local orbital forcing., Nature, 500(7463), 440–4, doi:10.1038/nature12376, 2013. Weller, R.: Postdepositional losses of methane sulfonate, nitrate, and chloride at the European Project for Ice Coring in Antarctica deep-drilling site in Dronning Maud Land, Antarctica, J. Geophys. Res., 109(D7), 1–9, doi:10.1029/2003JD004189, 2004. Wolff, E. W., Rankin, A. M. and Röthlisberger, R.: An ice core indicator of Antarctic sea ice production?, Geophys. Res. Lett., 30(22), 2–5, doi:10.1029/2003GL018454, 2003. Wolff, E. W., Fischer, H., Fundel, F., Ruth, U., Twarloh, B., Littot, G. C., Mulvaney, R., Röthlisberger, R., de Angelis, M., Boutron, C. F., Hansson, M., Jonsell, U., Hutterli, M. a, Lambert, F., Kaufmann, P., Stauffer, B., Stocker, T. F., Steffensen, J. P., Bigler, M., Siggaard-Andersen, M. L., Udisti, R., Becagli, S., Castellano, E., Severi, M., Wagenbach, D., Barbante, C., Gabrielli, P. and Gaspari, V.: Southern Ocean sea-ice extent, productivity and iron flux over the past eight glacial cycles., Nature, 440(7083), 491–496, doi:10.1038/nature06271, 2006.
* * *
[Figure]

[Figure]

**Fig. 1.**

[Figure]

**Fig. 2.**

[Figure]

Fig. 3.

**Summit-2010**
**Total Br and Br Enrichment (relative to Na)**

**Fig. 4.**

[Figure]

**Fig. 5.**

**Summit-2010**

a

Summit-2010 Annual Br vs MSA 1750-1880 C.E.

**Tunu**

b

Tunu Annual Br vs MSA 1750-1880 C.E.

c

Fit of Summit-2010 MSA used to calculate preindustrial MSA, Br relationship

d

Fit of Tunu MSA used to calculate preindustrial MSA, Br relationship

e

Summit-2010 Total Bromine and Sea Ice Bromine ( extrapolated from Br, MSA 1750-1880 relationship)

f

Tunu Total Bromine and Sea Ice Bromine ( extrapolated from Br, MSA 1750-1880 relationship)

**Fig. 6.**

**Fig. 7.**

[Figure]

**Fig. 8.**

---

## Author Response (AR1)

**Editor Decision: Publish subject to minor revisions (review by Editor)** (22 Aug 2016) by
Dr Kumiko Goto-Azuma
Comments to the Author:
Dear Dr. Maselli,

**I think you have addressed most of the referee comments. But some of them have not been fully addressed. I have a few comments for further revisions.**

**Q1:     As Referee #2 (Ref2 hereafter) commented, using the MSA data to determine the sea ice component of the Br signal is tricky. The substantial negative values of the non-sea-ice Br (Fig. 8) worries me. Can you give any reasonable explanation for the negative values? The large scatter of the data shown in Fig. R6a,b does not look very convincing, either, even though the correlation may be statistically significant. When you did a linear fitting between the MSA and Br data, you used smoothed data only for MSA. Wouldn't it be better if you use smoothed data also for Br?**

R1:     The negative values appear during the period when the non-sea ice Br is, on average, zero. The negative values are thus a result of the total Br being less than the smoothed MSA value. Though the sources of Br and MSA are linked – which is what provides the similarities between the general low frequency trend of the two species, the atmospheric processing, transport and deposition of the two species may be modified by different variables such as changes in aerosol acidity, for example. These variables cause the short term differences between the MSA and total Br records preserved in the ice so we believe it is not unreasonable to expect negative values in the calculated non-sea ice Br record when the MSA and total Br are close (essentially no non-sea ice Br).
Only the MSA data was smoothed (as opposed to Br and MSA both being smoothed) to try and minimize the amount of data manipulation in the production of the non-sea ice Br record. However, as can been seen from the Figure Q1 (attached, Summit-2010) a smoothing of the calculated non-sea ice Br record (as shown in the manuscript) produces the same result as if the non-sea ice Br record was calculated from the difference between the smoothed total Br record and the smoothed MSA record. Looking at the smoothed version of the non-sea ice Br record, it is clear that there is a sustained period (1815-1870 C.E.) over which the non-sea ice Br record is on average, below zero. In both the Summit and Tunu ice core sites this period corresponds to a period of significant volcanic activity (including the Tambora eruption of 1815 C.E.)  and this may be evidence acid induced loss of total Br, or perhaps a modification on the atmospheric MSA production by the increased sulfate levels.  This conclusion is pure speculation; however, it is supported by the observation that the other period of negative non-sea ice Br values in the Tunu record occurring during the period of elevated sulfate levels (~1860-1940 C.E.).
A comment has been added to the manuscript discussing the negative non-sea ice Br values and the possible links to the elevated sulfate values in section 4.2.2.

[Figure]

Figure Q1: Summit-2010 non-sea ice bromine (nsiBr; red line) calculated as the difference between the smoothed total bromine record (thick blue line) and the sea ice bromine record interpolated from the smoothed MSA record. For comparison the nsiBr record (thin purple line,as displayed in the manuscript) calculated from the difference between the annual Br (thin blue line) and the sea ice bromine record interpolated from the smoothed MSA record is also shown. The smoothing of the manuscript nsiBr is also shown (thick black line).

**Q2. Both Ref1 and Ref2 commented on biogenic sources of Br. I was also puzzled by the argument on biogenic sources of Br. To me, it is not very clear if the contribution of biogenic sources is important or not. Does the manuscript mean that the sea ice source is dominant in spring and biogenic sources are dominant in summer? Not having read the revised manuscript, I could not evaluate if the manuscript has been revised adequately. Please revise the manuscript carefully to make the logic clear.**

R2. This following text was added to the manuscript in Section 4.1.4. It aims to make clear that the biological sources are not thought to be the major bromine source for the ice core record. "The season of Arctic sea ice algae productivity is confined by limitations in available sunlight and nutrients resulting in a mid-to-late spring maxima – depending upon site location (Leu et al., 2015) – as is reflected in the seasonality of the MSA record. Direct transport of bromine enriched aerosols from these algal sources to the ice core sites again cannot explain the summer maximum of bromine observed in the ice. In addition to the incoherence of the seasonality of the bromine ice core signal, to-date biogenic sources have been considered insignificant sources of bromine in the Arctic marine boundary layer compared with the inorganic bromine source from sea salts (Simpson et al., 2007). "

**Q3. The " 3 step linear regression" is explained in author's response to Ref1. Please explain it also in the revised manuscript.**

R3. The explanation has been included in section 3.1

**Q4. Ref1 suggests that a more detailed consideration of potential impacts of acidity on the stability of Br and MSA in ice cores is needed, this has not been fully addressed. Though the authors argue that they analyzed the discrete samples obtained from the**

**melt stream, this does not address the Ref1's concern about a possibility that MSA might get destabilized at the melt head. Potential impacts of acidity variability within each ice core should be discussed, too, to fully address the Ref1 comment.**

R4.      The most conclusive way to determine whether there is destabilization of MSA at the CFA melter head would be to perform discrete MSA analyses on subsamples of the unmelted ice core directly and compare this with the discrete samples obtained at the end of the melt stream. However, this was not performed for this study. Instead we use the comparison between the MSA record generated from the Legrand GRIP ice core (collected only 35 km from the Summit-2010 site) to demonstrate that both techniques measured the same MSA concentrations which suggests no MSA is lost at the melt head. The same conclusion can be reached by comparison to the results of the Summit snow pit study by Jaffrezo et al., (1994). Over the period of winter 1984-1992 Jaffrezo measured MSA concentrations between ~1 and 5 ppb, with a mean of 2.1±1.8(1σ) ppb (calculated from a digitized version of the snowpit data). The annual measurements on the Summit-2010 ice core averaged 2.0±0.7 (1σ) ppb over the 1984-1992 period; comparable concentrations to the Jaffrezo study albeit with a lower variability. The comparison between the Legrand, Jaffrezo and Summit-2010 ice core MSA results have been discussed in the Manuscript in section 3.2 and the text box of Figure S4.

Also, it unlikely that MSA is 'destabilized' at the CFA melter head for several reasons;
-MSA is very chemically stable, only reacting with highly oxidizing species such as the OH radical.
-No OH radicals are produced at the melter head. $H_2O_2$ records from ice cores have been successfully measured using the continuous melt technique. If the OH radicals were produced at the melter head, then measurements of $H_2O_2$ in the ice cores would not be possible.
-The CFA melter head is made of chemically inert Silicon Carbide, so there are unlikely to be any surface catalyzed reactions.
- The melter head is only heated to ambient temperatures so no thermal degradation of the MSA is expected.

 The acidic variability within the ice cores is unlikely to affect the measurement of the MSA ice core record for several reasons:
-As discussed in the manuscript, the snow accumulation is large enough at both ice core sites that the amount of (acid induced) post-depositional loss of MSA is negligible.
- The MSA measurements were made on ice from the center of the ice core. No MSA is expected to be lost from the center of the ice cores during storage as there is no atmospheric exposure to the central ice.
-The internal isotopomer standard used as part of the MSA measurements would correct for any changes in instrument response due to variations in water chemistry.

**EC1: In Section 2.1, Fig. 1 needs to be referred to.**

REC1: reference to figure 1 has been included in Section 2.1

**EC2. The order of the figures or figure numbers should be re-organized. For example, Fig. 3 is referred to before Fig.2, and Fig.S3 before S1 and S2 (line 173) . Please check the figure numbers and the order that they are referred to.**

REC2. The figures have either been relabeled or the order to which they are referred in the text corrected.

**EC3. The legend for the blue color in Fig.8a reads exPb (x2 after 1925). Does this mean "after 1940?"**

REC3. Figure 8 has been modified for clarity– now there are only the 'exPb' and 'Br from leaded fuel' time-series. In the figure text box it explains that the 'Br from leaded fuel' time-series is created by assuming a 1:1 ratio between exPb and Br from fuel before 1925 and 1:2 ratio after 1925.

**EC4. Line 923. Delete one of the two "significant"s.**

REC4: Correction made

**EC5. Figure 5. Pink circles and the fonts beside the color bar are too small to see. Same applies to some of the other figures and supplementary figures. In some of the figures, legends, axis labels etc. are too small to see. Please make them larger.**

REC5. Figure 5 and S8 circles and color bar fonts increased in size. Figs 6,7 font size increased. All newly added plots have been checked for font size.

References:

[revised manuscript text omitted]

Page 47: [4] Deleted        olivia maselli        25/08/2016 9:04 PM

[Figure]

**Supplementary figures and tables**

**Table S1.** Summary of timings of each inflection in the 3-step linear regression of annual bromine and MSA at Summit and Tunu. Regression was performed on the data sets with outliers removed as described in Fig. 2. The signs indicate the direction of the inflection in the record, errors are 2σ.

**Timing of inflection (Year, C.E.)**

|  | Infl. 1 | | Infl. 2 | | Infl. 3 | | Infl. 4 |
|---|---|---|---|---|---|---|---|
|  | Br | MSA | Br | MSA | Br | MSA | Br |
| Summit-2010 | (-)1819 ±22 | (-)1854 ±12 | (+)1879 ±22 | (+)1878 ±12 | (+)1932 ±10 | (-)1930 ±16 | (-)1974 ±20 |
| Tunu | (-)1842 ±22 | (-)1812 ±12 | (+)1857 ±24 | (+)1821 ±21 | (+)1944 ±18 | (-)1984 ±4 | (+)1966 ±20 |

**Table S2.** Summary of the average aerosol concentrations as determined by the 3-step linear regression of annual bromine and MSA at Summit and Tunu displayed in Fig. 2. The duration of each step in concentration is bracketed by the inflection points summarized in Table S1. Concentrations are in units of nM. MSA did not show a stable period after the third infection in the series and so was not assigned a concentration value for 'Step 3'. Errors represent $2\sigma$ in the concentration value.

|  | **Concentration (nM)** | | | | |
|---|---|---|---|---|---|
|  | Step 1 | | Step 2 | | Step 3 |
|  | Br | MSA | Br | MSA | Br |
| Summit-2010 | 5.4±0.2 | 48±1 | 4.2±0.2 | 36±2 | 5.5±0.3 |
| Tunu | 4.2±0.3 | 25±1 | 3.2±0.3 | 21.2±0.7 | 4.8±0.5 |

[Figure]

**Figure S1.** Illustration of the shift in the seasonal MSA peak along the length of the Summit-2010 ice core. The difference in amplitude between the spring/summer and winter/fall MSA signal each year was calculated ((MAM+JJA)-(DJF+SON)) and observed to shift linearly along the length of the ice core. At the shallowest part of the ice core the positive values show the MSA peak appears in the spring/summer whilst in the deepest and oldest part of the ice core the signal has shifted to a winter/fall annual maximum. This phenomenon has previously been attributed to annual salt gradients within the ice core driving the migration of the MSA toward the higher salt location, winter (Mulvaney et al., 1992; Weller, 2004).

[Figure]

**Moved down [1]: Figure S1.**  ... [2]

**Moved down [5]: Figure S3.**

**Moved (insertion) [1]**

**Moved down [3]: Figure S2.**

**Moved down [4]:** $([Br]/[Na])_{seawater} = 1.793 \times 10^{-3}$, $([Br]/[Cl])_{seawater} = 1.539 \times 10^{-3}$.

**Moved down [6]: Figure S4.**

[Figure]

**Figure S2.** Comparison between the annual cycle in inorganic Br measured at Summit from snow samples taken as part of the GEOSummit project (2007-2013) and in the Summit-2010 ice core (1900-2010). The snow samples were analysed for inorganic Br on the same system used to measure the ice core records. The results of the snow samples support the observation from the ice cores that the maximum flux of Br is in summer with a possible secondary peak in spring. The error bars represent 1σ.

Moved (insertion) [3]

Moved (insertion) [2]

[Figure]

**Figure S3.** Monthly values of bromine, sodium and chlorine compared with their sea water ratio (red). At both sites, both the Br/Na and Br/Cl lie predominantly above the sea water ratio, whilst Cl/Na shows only a small Cl enrichment which increases at small sodium concentrations. At Tunu, 11% and 12% of the points show bromine depletion relative to Na and Cl, respectively. $([Br]/[Na])_{seawater} = 1.793 \times 10^{-3}$, $([Br]/[Cl])_{seawater} = 1.539 \times 10^{-3}$, $([Cl]/[Na])_{seawater} = 1.165$

Moved (insertion) [5]

Moved (insertion) [4]

[Figure]

**Figure S4.** Total bromine and bromine enrichment (relative to sodium) from the Summit-2010 ice core. The time-series have been plotted to match the signal variability in the preindustrial era (1750-1850 C.E.).

[Figure]

**Figure S5.** Comparison between the MSA record obtained from the GRIP ice core (Legrand et al., 1997) in 1993 and the Summit-2010 ice core from this study. The Summit-2010 ice core drill-site (72°20'N 38°17'24"W) is located 35 km SW of the GRIP ice core drill-site (72°34'N, 37°38'W). The GRIP MSA was measured in discrete samples using ion chromatography compared with the Summit-2010 ice core which was measured using the new technique of continuous melting of the ice core combined with continuous analysis by electrospray triple-quad mass spectrometry (as described in the text). The tight overlap between low frequency trend of the two series demonstrates that the new, continuous measurement technique is able to achieve a comparable accuracy in MSA concentration measurements to the discrete technique. It also demonstrates that negligible amounts of MSA are being lost during the continuous melt method. Discrepancies between the high frequency features of the two records is expected as the measurement resolution of the continuous method is much higher than the discrete method and the two records are from different ice core sites.

[Figure]

**Figure S6.** Demonstration of the reproducibility of the MSA online, continuous measurements performed on the Tunu ice core. Two different depths of the Tunu ice core are shown where the replicate analysis was performed by melting a secondary stick of ice cut from the same ice core and overlapping in depth: (a) Six 'overlap' ice cores were melted sequentially to replicate the MSA record over the depth 8-14 m.(b) Two 'overlap' ice cores were melted sequentially over the depth 186.2-187.9 m. Zooming in on a small section of the record at each depth demonstrates that the high frequency signal is real (not noise) and well replicated by the continuous MSA technique.

[Figure]

**Figure S7.** Comparison between discrete and continuous, online measurements of MSA measurements from the Tunu ice core samples. The discrete samples were collected as the continuous measurements were performed by directing part of the sample stream into an auto-sampler collection system just before they entered the analyzer. The samples were then frozen and later measured using ion chromatographic separation and the ESI/MS/MS detection. In this plot the continuous data have been averaged over the same depth range covered by each discrete sample and then both series plotted as the average age over that depth range. Over the 1750-2012 period the Tunu discrete measurements were, on average, 7% higher than the online measurements (dashed lines indicate average values over the 1750-2012 period). Both the discrete and continuous samples experienced identical conditions from ice melt to collection so the reason for offset in measured concentration is likely due to differences in post-processing of the data.

[Figure]

**Figure S8.** Total column air mass back trajectories from the (a) Summit-2010 and (b) Tunu ice core sites over the period 2005-2013 C.E. Maps display the fraction of the total number of trajectory hours (~100000 hrs month$^{-1}$) spent within the total vertical column (under 10000 m). Back trajectories were allowed to travel for 10 days. New trajectories were started every 12 hours. Map grid resolution is 2°x2°. Ice core locations are shown by a pink circle. Maps show that air masses consistently arrive at Summit from the SE Greenland coast with a smaller contribution from the SW coast, consistent with the trajectories seen in the boundary layer (Fig. 5). Air masses consistently arrive at Tunu from the western Greenland coast with a smaller contribution from the SE.

[Figure]

**Figure S9.** Autocorrelation maps of SIC during (a) the extended era (1900−2012 C.E.) and (b) satellite era (1979−2012 C.E.). Monthly SIC values were compared with the average SIC record from the area which shows the high positive correlation to the Summit-2010 MSA record (outlined in black in Figs. 6a, 6b). There is clearly a negative correlation between sea ice on the east and west coast which is seen over both era from March through to May, but the relationship turns positive in June and July over the extended time period (1900−2012 C.E.)

[Figure]

[Figure]

**Figure S10.** Analysis of the effect of errors in the ice core timescales on the correlation between the site MSA record and the local sea ice concentrations (SIC). By shifting the dating of the MSA records to either extreme of the dating error estimate and replotting the SIC correlation plots (Figs. 6 and 7) it is clear the error in the dating of the MSA records does not affect the sign of the correlations displayed on the maps but can have an affect on the magnitude of the correlation found in different locations. This is likely a result of the peaks in the MSA record being shifted in or out of temporal coherence with peaks in SIC at the different locations.

[Figure]

**Figure S11:** Annual cycle of open water in the ice pack (OWIP) within the aerosol source regions designated in Figs. 6 and 7. The annual cycle has been averaged over the period 1900-2012. The satellite period 1979-2012 shows the same temporal variability in OWIP at both sites but at reduced OWIP values.

[Figure]

**Figure S12:** Summary of the technique used to determine nsiBr: the amount of Bromine in excess of what is expected from a purely sea ice source. (a,b) Blue dots, blue fit line: correlation plots between total bromine and total MSA in Summit-2010 and Tunu ice cores, respectively over the preindustrial period 1750-1880 C.E.. Red dots, yellow fit line: Correlation plots between total bromine and smoothed MSA time series shown in c and d. (c,d) Annual MSA record fit with 9th order polynomial. (e,f) Comparison between the total bromine record (black) and the bromine predicted from the smoothed MSA, Br linear relationship determined in a and b (blue) – the bromine from a purely sea ice source. The difference between the blue and black lines in panels e and f is the amount of bromine in excess of what is expected from a purely sea ice source (nsiBr; see Fig. 8).

[Figure]

**Figure S13:** Comparison between nitrate and bromine records at both ice core sites. The time-series have been plotted to match the signal variability in the preindustrial era (1750-1850 C.E.). The difference between the two time-series is most dramatic at the Summit-2010 site because the sea ice record changes most dramatically at this site – and sea ice is the underlying driver of the bromine record.

| Page 1: [1] Deleted | olivia maselli | 25/08/2016 10:07 PM |

| Page 3: [2] Moved to page 3 (Move #1) | olivia maselli | 25/08/2016 10:07 PM |

**Figure S1.**

| Page 3: [3] Deleted | olivia maselli | 25/08/2016 10:07 PM |

---

## Author Response (AR2)

**cp-2016-49   Submitted on 24 Apr 2016**
**Sea ice and pollution-modulated changes in Greenland ice core methanesulfonate and bromine**
**O. J. Maselli, N. J. Chellman, M. Grieman, L. Layman, J. R. McConnell, D. Pasteris, R. H. Rhodes, E. Saltzman, and M. Sigl**

**Author's response, Sept 2016**

1. The statements on the MSA and Br temporal trends do not seem to agree with Table S1, which is confusing. For example, though Lines 185-186 reads "Ice core Br levels at each site were stable until ~1830", Table S1 shows different inflection points for Summit and Tunu, neither of which is 1830. Another example is Lines 283-284, reading " the decrease in both MSA and bromine at both sites in the early 1800s". But Table 1 shows different inflection points for MSA and Br, and for Summit and Tunu. Inflection 1 for Summit MSA is 1854 and that for Tunu Br is 1842. I don't think 1854 and 1842 can be called "early 1800s". Please revise the sentences describing the trends of MSA and Br to avoid these confusions. Also, please display the inflection points and straight regression lines in Fig. 2.

Lines 185- changed to:" Ice core Br levels at each site were stable until ~1820 at Summit and ~1840 at Tunu when they both decreased by ~1 nM, establishing a new baseline that was stable until the mid 1900s."

Lines 283 changed to :" the decrease in both MSA and bromine at both sites in the early to mid 1800s (Tables S1 and S2). In the 1900s, however, both sites show a divergence between the MSA and Br records"

The 3 step linear regression fit has been added to each data series in fig.2

2. The manuscript argues somewhat convincingly about MSA as a sea ice proxy, with the main argument based on a correlation between MSA and sea ice. But the correlation maps (Figures 6 and 7) are not very convincing. In black bordered regions both positive and negative correlations seem to be observed. But partly because the maps are too small, and colors for negative and positive correlations are hard to distinguish, it is hard to see in some regions if the correlation is positive or negative. I'm also confused by Fig. 6 because different black bordered regions are chosen for 1900-2010 and 1979-2010 without enough explanation. Please revise the correlation maps in Figs. 6 and 7 so that the correlations that the authors argue can be clearly seen, or maybe correlation maps can be removed.

Figures 6 and 7 have been edited so that only the SIC correlation maps from the months that show the best OWIP correlation have been kept. The retained maps have been enlarged. The original figure maps of SIC March-July have been placed in the supplementary. The manuscript focuses on the correlation between the MSA records and the total amount of open water in the ice pack within the black bordered regions which is an indication of the size of the marginal sea ice zone. As the editor notes it is hard to draw direct links between the SIC distribution within the black bordered region and the MSA records because the SIC varies so much within the bordered region that there are areas that show positive correlation and other areas which show negative correlation. The air masses that reach the ice core site take air from the whole black bordered region – in essence averaging out all the differences in SIC records. The SIC correlation maps are thus presented in order to demonstrate that this typical sea ice measure is not the best link to the MSA record - the OWIP in the pack is better.

The black bordered region of Fig. 6b has been reduced to be the same as 6a. The region was originally expanded because it increased the correlation of the OWIP record, but the increase was only modest and so for clarity the smaller region is used, and the increase in correlation with sample area only mentioned in the figure caption.

3. Please add enrBr(Na) and nsiBr to Figs. 6 and 7 together with OWIP and MSA. This would strengthen the arguments of the manuscript, if enrBr(Na) and nsiBr have very different trend from that of MSA in the recent period. Currently they are in the supplementary, but without OWIP trend in the same figure.
enrBr(Na) and nsiBr have been added to figs 6 and 7.

4. Please show enrBr(Na) and nsiBr in the main text, not in the supplementary, because enrBr is a sea ice proxy published by previous studies, and to strengthen the conclusion on the findings between enrBr/nsiBr and sea ice trends.
enrBr(Na) and nsiBr have been added to figs 6 and 7 as per #3.

5. It would be better to add a few words to the final line of the manuscript "leaving room for the possibility that bromine may still be an effective proxy for local Antarctic sea ice conditions" "*and for preindustrial sea ice reconstructions*."
This has been added to the final line.

Non-public comments to the Author:
Dear Dr. Maselli,

I have the following minor editorial comments. Please revise the manuscript following them.

1. Line 82: ~0.22 m yr-1. This needs a reference
This value was determined from the ice core record. This has been stated in the manuscript
2. Line 86: ~0.11 m yr-1. This needs a reference.
This value was determined from the ice core record. This has been stated in the manuscript
3.Line 114: a peristaltic pump. Information on the make and company is necessary.
Reference to the pump was removed
4. Line 117: M6 pump. What pump is this? A peristaltic pump?
Syringe-free liquid handling pump – this was added to the manuscript
5. Lines 124-125. "The analysis… (Saltzman et al., 2006)" should be moved to Line 116.
Done
6. Lines 135- 148. Please define suffixes obs, dust, seaweater and enrich.
done
7. Line 176. Please explain how outliers were removed.
Technique references Sigl 2013. Sentence reworded to remove the ambiguity.

8. In Methods section, please show how you define each month for the ice core data.
Original wording :

'The Summit-2010 and Tunu cores were dated using well-known volcanic horizons in sulfur (S). The dating of Summit-2010 was refined by annual layer counting using seasonal cycles in the ratio of non-sea salt S/Na (Sigl et al., 2015).'

Changed to :
'The Summit-2010 and Tunu cores were dated using volcanic horizons in sulfur (S) from well dated historic eruptions (e.g., 1815, 1835, 1846, 1854, 1873, 1883, 1912). The dating of both cores was refined by annual layer counting using seasonal cycles in Na, Ca, and the ratio of non-sea salt S/Na as described in more detail for another Greenland ice core (NEEM-2011-S1) by Sigl et al., (2013, 2015). Annual-layer boundaries (nominal January) were defined as the minimum value in the ratio of non-sea salt S/Na following Sigl et al. (2013). The seasonal cycles in Na and Ca (from sea-salt and mineral dust emissions peaking in winter months) remain largely unaffected by rising anthropogenic emissions during the industrial period and thus can be used for annual layer counting for the entire record. The minimum in hydrogen peroxide was also used as a winter marker in the upper section of the Summit-2010 core. Timing was evaluated for consistency against other parameters including insoluble particle counts and black carbon. Monthly values were calculated assuming a constant distribution of snowfall within each year. Because of the lower accumulation rate and strong katabatic winds at the Tunu site, constraints from volcanic synchronization played a more important role in the developing the depth-age scale for the Tunu core compared with Summit-2010. First the Tunu non-sea salt S record was synchronized to the NEEM-2011-S1 volcanic record (Sigl et al., 2015) and then the required number of annual layers between volcanic horizons picked from the high-resolution chemistry.
The annual-layer dating for these ice cores resulted in a plutonium record that is consistent with other ice cores from Greenland between 1950 and 1970 and with the emission histories from nuclear weapon testing in the Northern Hemisphere (Arienzo et al., 2016). The error in the dating of the ice core records was estimated as ± 0.33 years for the Summit-2010 record and ± 1 years for the Tunu record.

9. Lines 298-299. I think H2SO4 and HNO3 could be formed not only after deposition on the ice/snow, but also during transportation.
I agree – included adsorption ontp aerosols:
'$SO_2$ and $NO_x$ from the haze are adsorbed onto aerosols or deposited directly on the ice/snow and oxidised to sulfuric ($H_2SO_4$) and nitric acid ($HNO_3$)'

10. Lines 319-320: I think coal burning is one kind of fossil fuel combustion.
Changed to:
global $SO_2$ emissions with maxima from coal (~1920 C.E.) and coal plus petroleum combustion (~1970 C.E.),

11. Line 340: at heights of 500m and 10,000 m. I don't think the latter is correct. The latter should be total column trajectory
changed to:
"calculated for Summit-2010 up to heights of 500 and 10,000m (total column trajectory, Fig. 5a, S8a)"

12. Line 371, OWIP is stable until ~1970. To my eyes, OWIP seems to have started to decrease earlier than 1970 at Summit.
Changed to:

"For both ice cores the source region OWIP trend is followed by the MSA."

13. Line 473: Mirabolite should be mirabilite.
Done

14. Line 488: was less than 1nM. At Tunu, the exPb peak is close to 2nM (only slightly lower than 2nM). If this is corrected, 0.67M in Line 490 needs to be corrected, too.
The 1 nM reference is to the coal burning era – which peaks in 1920. The plots are consistent with this statement. At the height of the fossil fuel burning era ( ~1970) the exPb peak close to 2nM as the editor notes.

15. Line 531. Please remove "," between "reactive" and "species".
Done

16. Please check the references more carefully. I found the following errors, though I haven't checked the references very carefully.

Thanks, the reference list needed to be refereshed.
- Jaffrezo et al. (1994) (Line 258) is missing from the reference list. Updated
added
- Line 446 Macias Fauria et al (2010): Is Macias Fauria a family name? Though this is consistent with the reference list, I wonder if this is correct.
 Yes it is correct

- Line 483 McConnell et al. (2008, 2007):McConnell et al. (2008) is missing from the reference list. This was Mcconnell and Edwards 2008. Ref removed.

Line 809, Science (80-): Something missing? fixed
Line 615, Morin (2008): Is this Morin et al. (2008)? Updated

Figs. 5 and S8. In some of the maps, "month" can't be seen. Month labels updated

Caption of Fig. S4. I couldn't understand the meaning of the sentence "The time-series… (C.E.). This sentence has been removed from fig s4 and s15, it is not essential.

Caption of Fig.S6. 'overlap' ice cores. Does 'ice cores" mean "ice sticks" or "ice pieces"?
Changed to: Two different depths of the Tunu ice core are shown where the replicate analysis was performed by melting a secondary stick of ice cut from the same ice core and overlapping in depth ('overlap' ice sticks) : (a) Six 'overlap' ice sticks were melted sequentially to replicate the MSA record over the depth 8-14 m.(b) Two 'overlap' ice sticks were melted sequentially over the depth 186.2-187.9 m.

Fig. S10. The maps are too small to see. Maps enlarged

1. Please state in more detail how you dated the cores, and how you defined a year and months.
   - As for the Summit core, annual layer counting seems to have been done using nssS/Na ratio (Lines 88-89), and the manuscript refers to Sigl et al. (2015). But Sigl et al (2015) seems to have used not only nssS/Na ratio but also other parameters as well. If you used only nssS/Na, how did you deal with the change of seasonality in nssS due to anthropogenic
S input?

This has been address in #8.
-Was the Tunu core dated only by volcanic sulfur horizons?
This has been address in #8.

2.  Please add the average seasonal cycle of water stable isotopes
    to Fig. 3.
    The d18O annual cycle averaged over the 1900-2010 period has been added to the figure 3. And
    reference to the technique added to the manuscript. The record earlier than 1900 was not included as
    the record has not been corrected for back diffusion.

[revised manuscript text omitted]
, whilst Cl/Na shows only a small Cl enrichment which increases at small sodium concentrations. At Tunu, 11% and 12% of the points show bromine depletion relative to Na and Cl, respectively. $([Br]/[Na])_{seawater} = 1.793 \times 10^{-3}$, $([Br]/[Cl])_{seawater} = 1.539 \times 10^{-3}$. $([Cl]/[Na])_{seawater} = 1.165$

[Figure]

**Figure S4.** Total bromine and bromine enrichment (relative to sodium) from the Summit-2010 ice core.

[Figure]

**Figure S5.** Comparison between the MSA record obtained from the GRIP ice core (Legrand et al., 1997) in 1993 and the Summit-2010 ice core from this study. The Summit-2010 ice core drill-site (72°20'N 38°17'24"W) is located 35 km SW of the GRIP ice core drill-site (72°34'N, 37°38'W). The GRIP MSA was measured in discrete samples using ion chromatography compared with the Summit-2010 ice core which was measured using the new technique of continuous melting of the ice core combined with continuous analysis by electrospray triple-quad mass spectrometry (as described in the text). The tight overlap between low frequency trend of the two series demonstrates that the new, continuous measurement technique is able to achieve a comparable accuracy in MSA concentration measurements to the discrete technique. It also demonstrates that negligible amounts of MSA are being lost during the continuous melt method. Discrepancies between the high frequency features of the two records is expected as the measurement resolution of the continuous method is much higher than the discrete method and the two records are from different ice core sites.

[Figure]

[Figure]

**Figure S6.** Demonstration of the reproducibility of the MSA online, continuous measurements performed on the Tunu ice core. Two different depths of the Tunu ice core are shown where the replicate analysis was performed by melting a secondary stick of ice cut from the same ice core and overlapping in depth ('overlap' ice sticks) : (a) Six 'overlap' ice sticks were melted sequentially to replicate the MSA record over the depth 8-14 m.(b) Two 'overlap' ice sticks were melted sequentially over the depth 186.2-187.9 m. Zooming in on a small section of the record at each depth demonstrates that the high frequency signal is real (not noise) and well replicated by the continuous MSA technique.

| Deleted: cores |
| Deleted: cores |

[Figure]

**Figure S7.** Comparison between discrete and continuous, online measurements of MSA measurements from the Tunu ice core samples. The discrete samples were collected as the continuous measurements were performed by directing part of the sample stream into an auto-sampler collection system just before they entered the analyzer. The samples were then frozen and later measured using ion chromatographic separation and the ESI/MS/MS detection. In this plot the continuous data have been averaged over the same depth range covered by each discrete sample and then both series plotted as the average age over that depth range. Over the 1750-2012 period the Tunu discrete measurements were, on average, 7% higher than the online measurements (dashed lines indicate average values over the 1750-2012 period). Both the discrete and continuous samples experienced identical conditions from ice melt to collection so the reason for offset in measured concentration is likely due to differences in post-processing of the data.

[Figure]

[Figure]

**Figure S8.** Total column air mass back trajectories from the (a) Summit-2010 and (b) Tunu ice core sites over the period 2005-2013 C.E. Maps display the fraction of the total number of trajectory hours (~100000 hrs month⁻¹) spent within the total vertical column (under 10000 m). Back trajectories were allowed to travel for 10 days. New trajectories were started every 12 hours. Map grid resolution is 2°x2°. Ice core locations are shown by a pink circle. Maps show that air masses consistently arrive at Summit from the SE Greenland coast with a smaller contribution from the SW coast, consistent with the trajectories seen in the boundary layer (Fig. 5). Air masses consistently arrive at Tunu from the western Greenland coast with a smaller contribution from the SE.

**Summit-2010**

**(a)**            **1900-2010**

[Figure]

**(b)**            **1979-2010**

[Figure]

**Figure S9.** Correlation maps of monthly sea ice concentration (SIC) derived from the Summit-2010 ice core. (a) HadISST1 ICE dataset from 1900-2010 C.E. correlated with annual records of MSA. Outliers were removed from the MSA records before the correlations were performed to prevent distortion of the correlations. Month labels indicate the month of SIC compared with the annual MSA value. Only locations that showed a SIC variability greater than 10% and have a significant correlation (t-test, $p<0.05$) are displayed. The area of sea ice that is the likely source of MSA (as indicated by the air mass trajectories) are outlined in black [70°− 63°N, 0°− 45°W]. (b) As for (a) but focused on the satellite period 1979-2010 C.E. and the outlined area covers [70°−63°N, 0°−60°W].

**Tunu**

[Figure]

Figure S10. Correlation maps of monthly sea ice concentration (SIC) derived from the Tunu ice core. (a) HadISST1 ICE dataset from 1900-2012 C.E. correlated with annual records of MSA. Outliers were removed from the MSA records before the correlations were performed to prevent distortion of the correlations. Month labels indicate the month of SIC compared with the annual MSA value. Only locations that showed a SIC variability greater than 10% and have a significant correlation (t-test, $p<0.05$) are displayed. The area of sea ice that is the likely source of MSA (as indicated by the air mass trajectories) are outlined in black [77°–67°N, 62°–50°W]. (b) As for (a) but focused on the satellite period 1979-2012 C.E.

[Figure]

**Figure S11.** Autocorrelation maps of SIC during (a) the extended era (1900−2012 C.E.) and (b) satellite era (1979−2012 C.E.). Monthly SIC values were compared with the average SIC record from the area which shows the high positive correlation to the Summit-2010 MSA record (outlined in black in Figs. 6a, 6b). There is clearly a negative correlation between sea ice on the east and west coast which is seen over both era from

March through to May, but the relationship turns positive in June and July over the extended time period (1900−2012 C.E.)

[Figure]

Figure S12

[Figure]

Sea Ice Concentration correlated with Tunu MSA

Tunu MSA timescale shifted 1 year earlier

Tunu MSA timescale not shifted

Tunu MSA timescale shifted 1 year later

**Figure S12.** Analysis of the effect of errors in the ice core timescales on the correlation between the site MSA record and the local sea ice concentrations (SIC). By shifting the dating of the MSA records to either extreme of the dating error estimate and replotting the SIC correlation plots (Figs. 6 and 7) it is clear the error in the dating of the MSA records does not affect the sign of the correlations displayed on the maps but can have an affect on the magnitude of the correlation found in different locations. This is likely a result of the peaks in the MSA record being shifted in or out of temporal coherence with peaks in SIC at the different locations.

[Figure]

**Figure S13:** Annual cycle of open water in the ice pack (OWIP) within the aerosol source regions designated in Figs. 6 and 7. The annual cycle has been averaged over the period 1900-2012. The satellite period 1979-2012 shows the same temporal variability in OWIP at both sites but at reduced OWIP values.

[Figure]

[Figure]

**Figure S14:** Summary of the technique used to determine nsiBr: the amount of Bromine in excess of what is expected from a purely sea ice source. (a,b) Blue dots, blue fit line: correlation plots between total bromine and total MSA in Summit-2010 and Tunu ice cores, respectively over the preindustrial period 1750-1880 C.E..

Red dots, yellow fit line: Correlation plots between total bromine and smoothed MSA time series shown in c and d. (c,d) Annual MSA record fit with 9th order polynomial. (e,f) Comparison between the total bromine record (black) and the bromine predicted from the smoothed MSA, Br linear relationship determined in a and b (blue) – the bromine from a purely sea ice source. The difference between the blue and black lines in panels e and f is the amount of bromine in excess of what is expected from a purely sea ice source (nsiBr; see Fig.
8).

[Figure]

 **Figure S15:** Comparison between nitrate and bromine records at both ice core sites. The difference between

 the two time-series is most dramatic at the Summit-2010 site because the sea ice record changes most

 dramatically at this site – and sea ice is the underlying driver of the bromine record.

| Deleted: 3 |
| --- |
| **Deleted:** The time-series have been plotted to match the signal variability in the preindustrial era (1750-1850 C.E.). |

[Figure]

Fit of Summit-2010 MSA used to calculate
preindustrial MSA, Br relationship

[Figure]

Fit of Tunu MSA used to calculate
preindustrial MSA, Br relationship

---

## Author Response (AR3)

**Editor Decision: Publish subject to minor revisions (review by Editor)** (12 Oct 2016) by Dr Kumiko Goto-Azuma

Comments to the Author:

1. Section 2.1 and Fig. 3.

22 cm water eq and 12 months resolution, makes each month less than 2 cm. Is this really resolved all the way in the record at Summit (Fig 3)?

Yes, we typically achieved sub 1 cm resolution. Which means at least one data point per month.

How come there is a shift between the two time periods in Na, if Na is the main parameter used to make the annual layer counting?

The minimum in the ratio of S/Na is the parameter that is used to define the annual layer boundaries, so any shift in the pure Na signal must be compensated by shift in S in the opposite direction. Pure Na, Ca and  $H_2O_2$  signals maxima were identified in order to confirm the identification of an annual cycle.

2. To me, lines 532-535 (P. 22) seem to contradict with lines 540-542. Please clarify the logic.

I think the Editor is referring to lines 632 and 640 of page 22(?) As I can see there is an apparent contradiction between the two statements. It is very clear from the high resolution comparison between the Br and S records that there is no immediate response to the volcanic eruptions reflected in the bromine record. So I don't think that the second statement on line 640 is dispute. The earlier statement made a suggestion that the atmospheric sulfur may influence the nsiBr record and could be the cause of the negative values because there was some coarse overlap in the timing of volcanic activity and industrial activity and the negative nsiBr values. This is a weakly surmised conclusion, (and as noted by the Editor, is in contradiction to the later statement) it was purely speculative – there are many things that happen coincidently with broad periods of negative nsiBr values, it was merely an observation that sulfur increases were one such factor – so to avoid confusion the statement has been removed from the manuscript.

3. The caption of Fig. 6 (line 1073).

3-1. There seems to be no red curve in Fig.6. There are thin and thick orange curves instead.

The line has been made thicker and a brighter red. Caption updated

3-2. The legend for the thin orange curve reads "MSA no volc". Isn't this "MSA, outliers removed"? These mean the same thing but to be consistent the legend wording was changed to "outliers rem." Also the lines were labelled incorrectly. The red one is the raw MSA and the orange one "outlier rem."

4. Same comments for Fig. 7 caption.
updated as above
5. The last line of the Fig. S9 caption.
The longitude seems to be wrong. It should be same as that for (a).
Corrected thanks.

- 1 Sea ice and pollution-modulated changes in Greenland ice core
- 2 methanesulfonate and bromine
- 3 O.J. Maselli1\*, N.J. Chellman1, M. Grieman2, L. Layman1, J. R. McConnell1, D. Pasteris1,
  4 R.H. Rhodes3, E. Saltzman2, M. Sigl1

[revised manuscript text omitted]

- <del>)</del>99
- )00